# Climate change may outpace current wheat breeding yield improvements in North America

Tianyi Zhang [1,2,13] ✉, Yong He [3,13] ✉, Ron DePauw [4], Zhenong Jin [5], David Garvin [6], Xu Yue [7], Weston Anderson [8,9], Tao Li [10], Xin Dong [1], Tao Zhang [11] & Xiaoguang Yang [12] ✉

Variety adaptation to future climate for wheat is important but lacks comprehensive understanding. Here, we evaluate genetic advancement under current and future climate using a dataset of wheat breeding nurseries in North America during 1960-2018. Results show that yields declined by 3.6% per 1 °C warming for advanced winter wheat breeding lines, compared with −5.5% for the check variety, indicating a superior climate-resilience. However, advanced spring wheat breeding lines showed a 7.5% yield reduction per 1 °C warming, which is more sensitive than a 7.1% reduction for the check variety, indicating climate resilience is not improved and may even decline for spring wheat. Under future climate of SSP scenarios, yields of winter and spring wheat exhibit declining trends even with advanced breeding lines, suggesting future climate warming could outpace the yield gains from current breeding progress. Our study highlights that the adaptation progress following the current wheat breeding strategies is challenging.

Wheat (*Triticum aestivum* L.) is among the world's most important staple food crops[1]. The Great Plains of North America is responsible for ~10% of annual global wheat production, and, more importantly, produces 30% of the world's high-quality wheat exports[2]. Wheat production in North America, therefore, represents a significant part of both regional food production and of the global food supply chain.

Over the last few decades, however, the rising temperature has been identified as a critical threat to wheat production in North America[3–5]. Without adaptation, climate change is anticipated to lower wheat yields in North America[6,7] by 1.0–10.0% per degree warming[8–10],

with associated weather extremes reducing yield stability[11]. Many plant species are negatively affected by high-temperature extremes, especially during the floral stage[12] of plant growth, when exposed to high temperatures can shorten the grain filling period[13] and consequently lower grain yields[3–6].

To counteract these negative effects, modeling studies recommend that farmers replace early-maturing varieties with late-maturating types to extend the length of the growing period, which would otherwise be shortened by warming[13,14]. Such a variety of adaptation assumptions has been widely used in modeling scenario

[1]State Key Laboratory of Atmospheric Boundary Layer Physics and Atmospheric Chemistry, Institute of Atmospheric Physics, Chinese Academy of Sciences, Beijing, China. [2]Collaborative Innovation Center on Forecast and Evaluation of Meteorological Disasters, Nanjing University of Information Science & Technology, Nanjing, China. [3]Institute of Environment and Sustainable Development in Agriculture, Chinese Academy of Agricultural Sciences, Beijing, China. [4]Advancing Wheat Technologies, 118 Strathcona Rd SW, Calgary, AB T3H 1P3, Canada. [5]Department of Bioproducts and Biosystems Engineering, University of Minnesota, St. Paul, MN, USA. [6]Formerly USDA-ARS Plant Science Research Unit, St. Paul, MN, USA. [7]School of Environmental Science and Engineering, Nanjing University of Information Science and Technology, Nanjing, China. [8]The International Research Institute for Climate and Society, Palisades, NY, USA. [9]Earth System Science Interdisciplinary Center, University of Maryland, College Park, MD, USA. [10]DNDC Applications, Research and Training, 87 Packers Falls Road, Durham, NH 03824, USA. [11]Institute of Microbiology, Chinese Academy of Sciences, Beijing, China. [12]College of Resources and Environmental Sciences, China Agricultural University, Beijing, China. [13]These authors contributed equally: Tianyi Zhang, Yong He. ✉e-mail: zhangty@mail.iap.ac.cn; heyong01@caas.cn; yangxg@cau.edu.cn

analysis[13,14] and identified as a promising strategy to adapt to future climate change stresses[15]. However, variety adaptation addressing only the length of the growing cycle is highly hypothetical and often does not fully capture the actual effects of variety adaptation in wheat. Therefore, identifying and benchmarking current breeding progress under historical climate trends could not only corroborate the modeling studies but also guide future breeding programs to better abate climate change stresses. To date, the scarcity of long-term field observations has hindered our understanding of how genetic advancements have affected the climate sensitivities of wheat yields.

Our study is based on the annual regional reports of the Northern Regional Performance Nursery (NRPN) and Southern Regional Performance Nursery (SRPN) for winter wheat, and Hard Red Spring Wheat Uniform Regional Nursery (HRSWURN) for spring wheat, with 85,770 data points (58,472 for winter wheat and 27,298 for spring wheat) in 92 sites over 1960–2018 in the Great Plains of North America (Fig. 1; see the "Methods" section). In the dataset, a common check variety and an annual set of advanced breeding lines were planted together in the same trial each season. The variety Kharkof was used as the long-term winter wheat check and the variety Marquis as the long-term spring wheat check. The two check varieties were planted in each site throughout the study period, which could be viewed as yield performance without variety replacement. On the other hand, the advanced breeding lines entered each year were different from previous year, which reflects the ongoing nature of wheat breeding itself over the study period. Therefore, the difference in yield sensitivities (advanced breeding lines vs. check variety) can be defined as the effectiveness of the historical variety development under climate change, providing a footprint of the actual wheat breeding effort. We analyze the panel data using a fixed-effect regression model (see the "Methods" section). Here, we focus on the time series of high-yielding genotype (HYG, i.e., the 97.5% percentile grain yield for advanced breeding lines each year), median-yielding genotype (MYG, i.e., 50% percentile), low-yielding genotype (LYG, i.e., 2.5% percentile), and the check variety (CK, i.e., yields of Kharkof for winter wheat and Marquis for spring wheat) (Supplementary Fig. 1). It should be noted that our dataset and method cannot be used to quantify the $CO_2$ fertilization effects and potential changes in breeding speed brought by new agricultural technologies.

## Results

### Benchmark the real-world breeding for winter and spring wheat

The effects of climate variations were assessed by regression of wheat yields on measures of cumulative exposure to freezing degree-day (FDD, °Cd, the cumulative temperature lower than the threshold that causes freezing injury), growing-degree-days (GDD, °Cd, the cumulative temperature between base and optimum growing temperature thresholds), extreme-growing-degree days (EDD, °Cd, the cumulative temperature above the optimum growing temperature threshold) and precipitation (Prcp) over growing season (see the "Methods" section). Separate regressions were carried out for CK, LYG, MYG, and HYG yield data, with 95% confidence intervals calculated by bootstrapping the site-years in the model by 1000 times.

Model regression coefficients suggest that effect of GDD is weak and statistically insignificant for both winter ($t = -0.3669$, $p = 0.7137$ for CK; $t = -0.641$, $p = 0.5216$ for MYG; $t = -1.032$, $p = 0.3022$ for HYG; $t = 0.178$, $p = 0.8585$ for LYG) and spring wheat yields ($t = 0.183$; $p = 0.854$ for CK; $t = 0.709$, $p = 0.478$ for MYG; $t = 0.957$, $p = 0.3386$ for HYG; $t = 0.064$, $p = 0.9486$ for LYG) under a two-tailed $t$-test, while yields were reduced significantly with higher EDD for winter ($t = -3.683$, $p = 0.000239$ for CK; $t = -3.202$, $p = 0.00139$ for MYG; $t = -3.291$, $p = 0.00102$ for HYG; $t = -3.137$, $p = 0.00173$ for LYG) and spring wheat ($t = -4.910$, $p = 1.12 \times 10^{-6}$ for CK; $t = -6.949$, $p = 8.03 \times 10^{-12}$ for MYG; $t = -7.242$, $p = 1.11 \times 10^{-12}$ for HYG; $t = -4.866$,

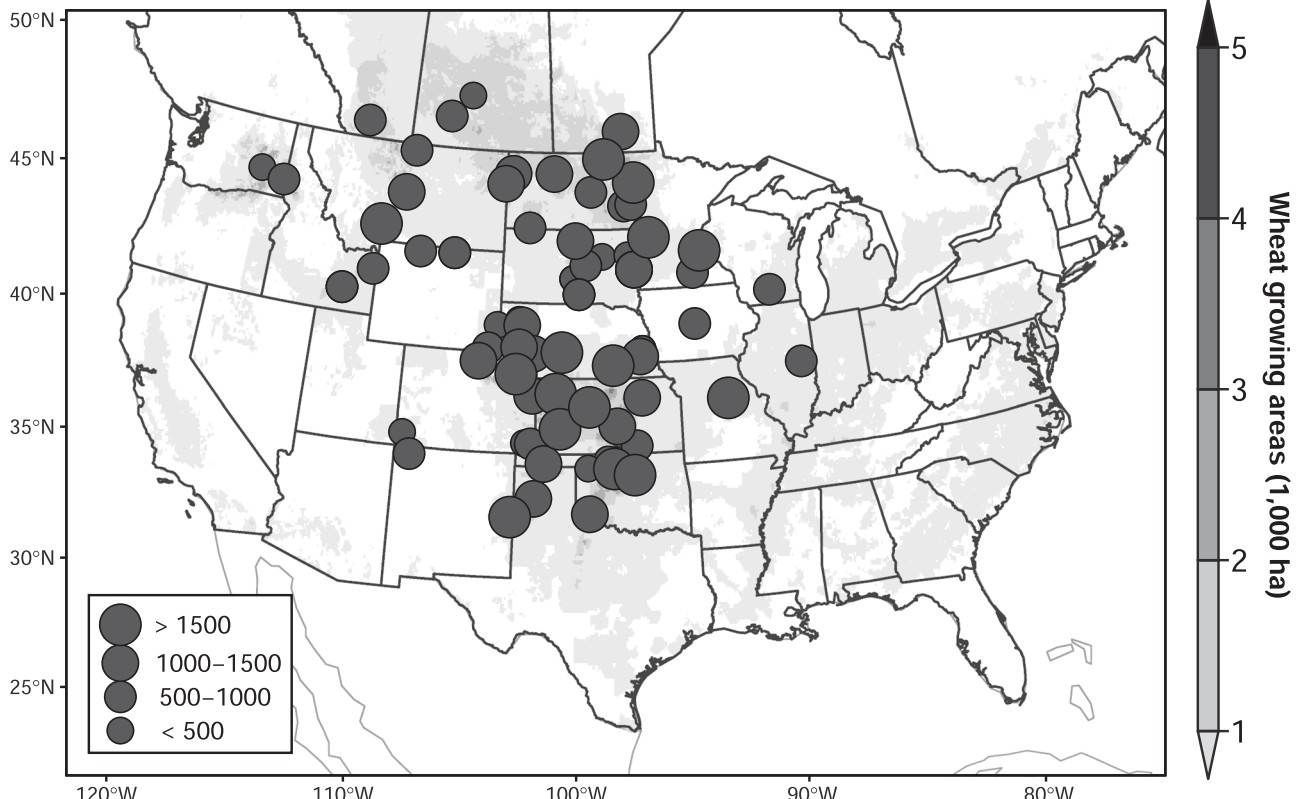

**Fig. 1 | The study region in North America.** The circles show locations of experimental sites, with the size of the circles indicating the number of data per site. The background map shows wheat-growing areas. Created by Tianyi Zhang using R version 3.6.1, under GNU General Public License v2, 1991 (gnu.org/licenses/old-licenses/gpl-2.0.en.html). Source data are provided as a Source Data file.

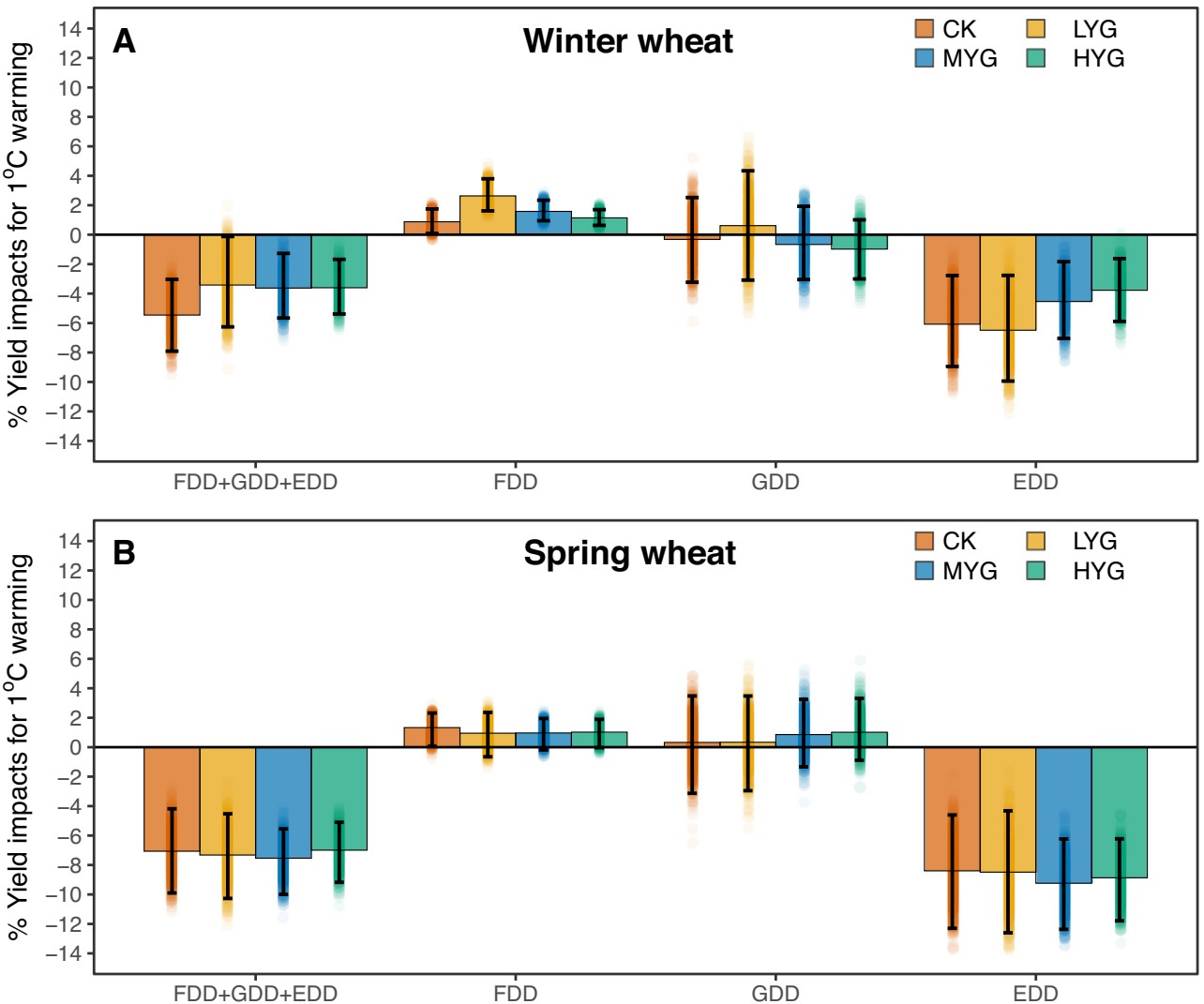

**Fig. 2 | Model estimate of yield impact of 1 °C warming on the four categories of genotypes of winter and spring wheat. A** Estimated yield changes induced by the net effect of 1 °C warming and the associated individual effects of freezing degree-day (FDD), growing-degree-day (GDD), extreme-growing-degree (EDD) in winter wheat. **B** Estimated yield changes induced by the net effect of 1 °C warming and the associated individual effects of FDD, GDD, EDD in spring wheat. The bar's color denotes genotypes: high-yielding genotype (HYG); median-yielding genotype (MYG); low-yielding genotype (LYG); check variety (CK). The centers of error bars denote median yield impact, and the error bars show the 95% confidence interval produced by bootstrapping method (n = 1000). The background data points are the yield impacts for each estimation by bootstrap analysis. Source data are provided as a Source Data file.

$p = 1.39 \times 10^{-6}$ for LYG) under a two-tailed $t$-test (Supplementary Table 1). A statistically insignificant effect of GDD and significant harmful impact of EDD are consistent with earlier findings[16]. An increase in the number of FDD negatively affects both winter ($t = -2.219$, $p = 0.0266$ for CK; $t = -4.401$, $p = 1.15 \times 10^{-5}$ for MYG; $t = -3.850$, $p = 1.23 \times 10^{-4}$ for HYG; $t = -4.934$, $p = 8.92 \times 10^{-7}$ for LYG) and spring wheat yields ($t = -3.079$, $p = 0.0021$ for CK; $t = -2.924$, $p = 0.0035$ for MYG; $t = -3.282$, $p = 0.00108$ for HYG; $t = -2.367$, $p = 0.0181$ for LYG) under a two-tailed $t$-test (Supplementary Table 1), which reflects the critical role of winter-killing[17] and freezing between stem elongation and grain maturation[18] in wheat production in North America.

To illustrate the impact of warming on yields, we estimate the net effect of 1 °C warming, as well as the individual effects of each temperature index, on yield for each genotype in a bootstrap analysis (see the "Methods" section) by artificially raising observed temperatures in each day by 1 °C for the entire crop growth cycle (Fig. 2). For winter wheat (Fig. 2A), there is 3.4–3.6% yield reduction per 1 °C warming for LYG, MYG, and HYG, which is less sensitive than the check variety

(−5.5% per 1 °C warming). This can be attributed to less yield losses due to increased EDD and greater yield benefits to decreased FDD for the advanced breeding lines relative to the check variety. For example, with 1 °C warming, advanced breeding lines are 1.6% less sensitive than check varieties to the associated increase in EDDs (MYG: −4.5% with −7.0% to −1.8% of 95% confidence interval versus CK: −6.1% with −9.0% to −2.8% of 95% confidence interval in a bootstrap analysis). One-degree warming also brings 0.70% greater yield benefits due to the associated decrease in FDD for MYG relative to CK (MYG: +1.58% with 0.95–2.34% of 95% confidence interval versus CK: +0.88% with 0.09–1.75% of 95% confidence interval in a bootstrap analysis). Yield response of HYG is similar, while results for LYG are slightly different, with greater yield sensitivity to EDD (−6.5% with −9.94 to −2.77% of 95% confidence interval) and to FDD (+2.6% with 1.62–3.80% of 95% confidence interval in a bootstrap analysis) when compared to CK (Fig. 2A).

Results of spring wheat differ from winter wheat. With 1 °C warming, variety replacement does not provide an advantage in improving climate resilience for spring wheat (Fig. 2B). Advanced

spring wheat breeding lines are slightly more sensitive to warming than check variety (Fig. 2B), with a 7.5% yield decline per 1 °C warming for MYG (−10.0% to −5.55% of 95% confidence interval) compared with a 7.1% yield decline per 1 °C warming for CK (−9.90% to −4.18% of 95% confidence interval in a bootstrap analysis). The reason is two folds; with 1 °C warming, increased EDD has a greater negative impact on advanced breeding lines of spring wheat when compared to check varieties (MYG: −9.2% with −12.4% to −6.2% of 95% confidence interval versus CK: −8.4% with −12.3% to −4.6% of 95% confidence interval in a bootstrap analysis), and the associated decrease in FDD is more beneficial to check varieties than it is too advanced breeding lines (MYG: +0.97% with −0.18 to +1.95 of 95% confidence interval versus CK: +1.33% with 0.06−2.32% of 95% confidence interval in a bootstrap analysis).

### Declining yield from breeding with future warming

With the above-quantified current breeding progress under historical climate, we then project future yield changes relative to the historical yield of check varieties due to climate change in four shared socioeconomic pathways (SSPs) (SSP1-2.6, SSP2-4.5, SSP3-7.0, SSP5-8.5) driven by six climate models in the Coupled Model Intercomparison Project Phase 6 (ACCESS-ESM1-5, BCC-CSM2-MR, CNRM-CM6-1, CNRM-ESM2-1, GFDL-ESM4, IPSL-CM6A-LR)[19]. Changes in the climate under the four SSPs scenarios are shown in Supplementary Fig. 2. In our SSPs, climate models project increasing GDD and EDD but decreasing FDD over time. Projections of precipitation are mixed in the direction of changes between models and show an unclear trend.

Projections using the check varieties in the future indicate that the yields would decrease to 12.7% in SSP1-2.6 (Fig. 3A) and 47.2% in SSP5-8.5 (Fig. 3G) for Kharkof at the end of the period, and the decreases for Marquis are even higher (24.8% in SSP1-2.6 of Fig. 3B, and 62.7% in SSP5-8.5 of Fig. 3H). Although MYG exhibited around 30% higher yields relative to CK in the baseline climate, such benefits would, in many cases, be overshadowed by the negative effects of warming over time (Fig. 3A, C, E, G). For winter wheat, the MYG yields show only moderate change in the SSP1-2.6 scenario over time (Fig. 3A), while more serious warming would result in a greater yield reduction in higher SSPs; e.g., the yield of MYG would reduce to the historical yield level of Kharkof with around 6 °C warming in SSP3-7.0 (Fig. 3E) and SSP5-8.5 (Fig. 3G). Spring wheat is more vulnerable than winter wheat. A mean growing season warming of 3.6 °C would reduce the yield of MYG back to the historical level of Marquis in the SSP2-4.5 (Fig. 3D), SSP3-7.0 (Fig. 3F), and SSP5-8.5 (Fig. 3H) scenarios.

Figure 4 represents yield gains of various genotypes relative to the check varieties during the corresponding period in the four SSPs. For winter wheat, by 2030s and 2050s, yield gains of MYG were maintained at around 46% on average (LYG: 3.4%; HYG: 84.0%) and 50% (LYG: 4.4%; HYG: 86.1%), respectively, with moderate variability among SSPs. By 2090s, the gain increases with warming SSPs, showing an average of 47.2% (LYG: 3.9%; HYG: 85.5%) benefit in SSP1-2.6 to 64.8% (LYG: 7.3%; HYG: 115.4%) in SSP5-8.5. For spring wheat, there is no substantial difference between 2030s and 2050s, with an average 44.9% gain. However, by 2090s, yield gains of MYG were projected to be reduced to 43.8% on average (LYG: 9.9%; HYG: 76.0%) in SSP1-2.6 and to 32.6% (LYG: 9.6%; HYG: 67.4%) in SSP5-8.5.

### Discussion

We provide an empirical assessment to benchmark the real-world effect of breeding on the climate sensitivity of wheat. Our results are based on long-term breeding nursery observations from multiple sites in which the climate exposures of advanced breeding lines and constant check varieties can be strictly controlled in the Great Plains of North America.

To detect the robustness of our results, we compared our regression results with earlier studies, finding that the estimated yield declines based on our nursery data fall within the range of earlier

assessments derived from other data sources[8,9,20] and using process-based modeling approaches[3,14,21] (Supplementary Table 2). Secondly, when the above statistical analysis was repeated using the county-level yield statistics, the yield responses of MYG to 1 °C warming derived here were close to the estimation from county-level yield statistics (Supplementary Fig. 3). This reflected that the response of MYG can approximate and represent the yield sensitivity in farmer's fields at a regional scale. Our results were also robust across a variety of panel regression model configurations. For example, we calculated the values of climate variables with varying time windows to define the growing season (Supplementary Fig. 4) and found a similar yield response to warming. We furthermore found that the model also produces similar results by including daily maximum temperature ($T_{max}$), a quadratic term for daily maximum temperature ($T_{max}^2$), daily average temperature ($T_{avg}$), and a quadratic term for daily average temperature ($T_{avg}^2$) in the entire growing season (Supplementary Fig. 5; Supplementary Table 3). Finally, we apply the temperature thresholds of spring wheat in Supplementary Table 4 to winter wheat from heading to maturity, and find similar results for winter wheat yield response to 1 °C warming (Supplementary Fig. 6) compared with the results presented in the main text (Fig. 2a). This suggests the different yield response between winter and spring wheat derived here is insensitive to temperature threshold settings in quantifying above temperature indices.

Empirical evidence shows that the effect of current breeding differs between winter and spring wheat, which has not been presented in previous studies. For spring wheat, historical breeding does not bring superior performance in climate resilience to the new varieties. Warming poses a more harmful effect on the yield of spring wheat breeding, resulting primarily from advanced spring wheat breeding lines having a greater yield sensitivity to high-temperature extremes than do the check varieties. This may be because the floral stage of spring wheat typically coincides with a higher temperature cycle than winter wheat. In contrast, advanced winter wheat breeding lines exhibit a greater ability to tolerate high-temperature extremes than the check variety, indicating greater climate adaptability. Such contrasting responses result in greater yield benefits for winter wheat than for spring wheat from breeding progress. Spring wheat is projected to suffer greater yield losses compared to winter wheat with future temperature increases. Therefore, we conclude that variety adaptation to climate warming is more challenging for spring wheat than winter wheat in North America. Such differences in responses show the need to understand further the underlying genetic process involved, such as specific vernalization requirements[22] and a variety of responses to heat/cold stress[23] between winter wheat versus spring wheat. More control field experiments are needed to examine the genetic basis across winter and spring wheat genotypes in future investigations.

It should be noted that our result is inconsistent with the overall effectiveness of a variety of adaptations derived in an earlier modeling study[13]. Despite great progress in crop modeling's ability to capture regional yield variability[3-5], the variety of adaptation scenarios used in these studies is less realistic. In their model settings, it was assumed that adapted varieties are late-maturing provided a suitable temperature regime, which is always available enabling the growing period length to be unchanged with warming[13]. However, maintaining the growing season length with increased temperature may oversimplify the need to adapt to climate warming. Additionally, Tack et al.[9] reported that more recently released winter wheat varieties were less able to resist high-temperature extremes than older varieties in Kansas, which seems to conflict with our assessment. However, the long-term constant check variety does not exist in their analysis[9]. The old and new varieties are grown under different years and climates, which may bias their yield responses because the growing environments, especially for temperature regimes, between old and new varieties were not strictly controlled. Conversely, in our dataset, adapted variety

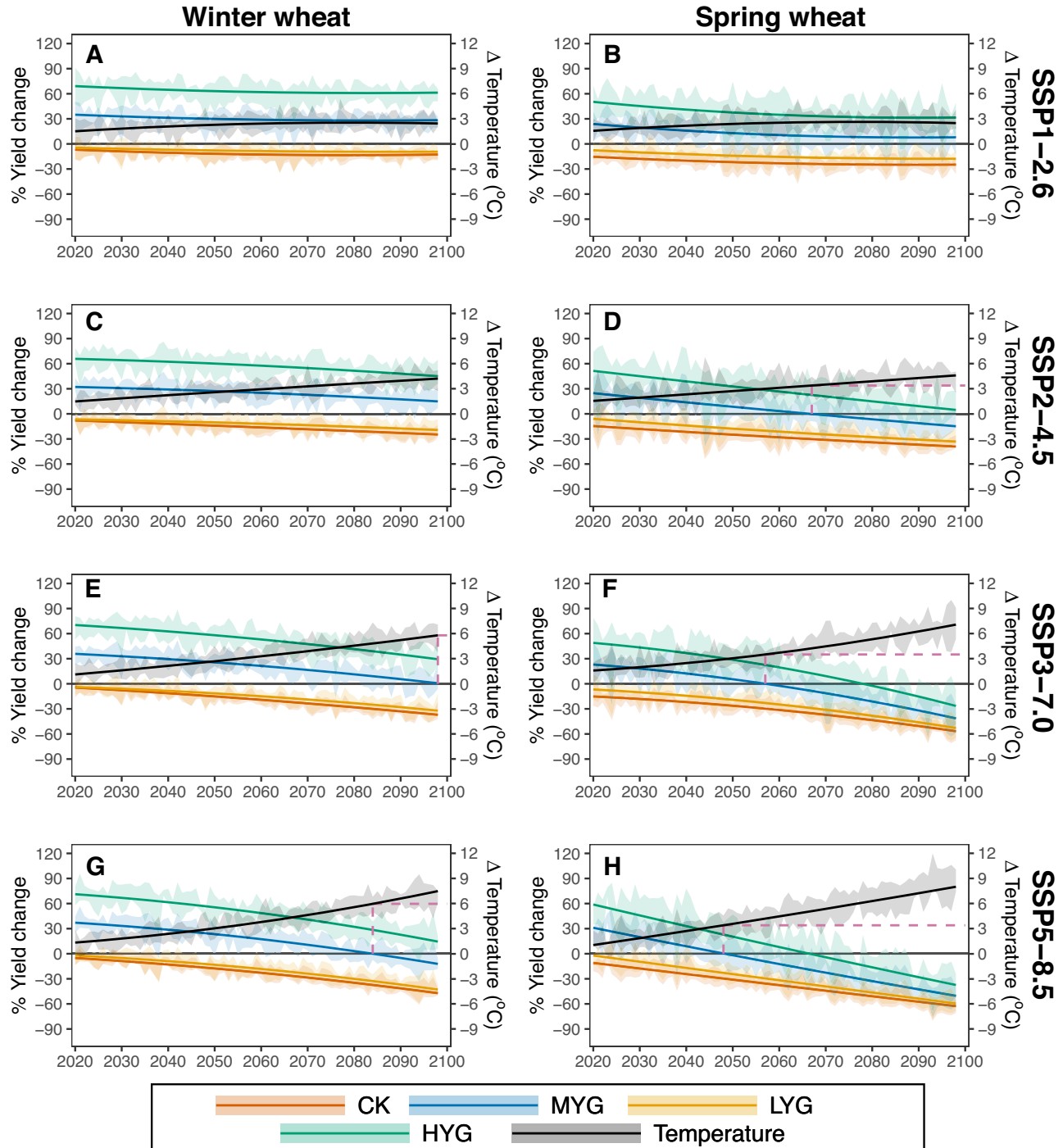

**Fig. 3 | Model projection of wheat yield changes due to changes in the climate of four categories of genotype under four SSPs, relative to the yield of CK at the baseline period.** The left panel shows results for winter wheat (**A**, **C**, **E**, **G**) and the right panel (**B**, **D**, **F**, **H**) shows spring wheat. The first row shows the result for the SSP1-2.6 scenario (A, B), the second row for SSP2-4.5 (**C**, **D**), the third row for SSP3-7.0 (**E**, **F**), and the fourth row for SSP5-8.5 (**G**, **H**). The color denotes genotypes: high-yielding genotype (HYG); median-yielding genotype (MYG); low-yielding genotype (LYG); check variety (CK) and temperature. The dashed line is the year and temperature that MYG projected to drop to the yield level of CK in the baseline. The lines are the best fit of the median changes in each year, and the shading areas show an estimate of the 95% confidence interval from six climate models. Source data are provided as a Source Data file.

and a constant check variety were always grown under the same year and overcome this difficulty from observations.

Our assessment suggests that the current breeding effort of adapting wheat to climate warming by tailoring the genotypes to the most likely long-term change remains insufficient in North America, which is not a rare example worldwide. Less satisfied wheat breeding progress in Western European[24] and Central Asian[25] countries has made their wheat subjected to a decline in climate resilience since

2000. A recent global-scale study[26] also pointed out that climate change may impede the rate of genetic gains in both spring wheat and durum wheat breeding. The plausible reason for the delayed variety adaptation progress is that the rate of climate change continues to outpace the overall process of breeding, delivery, and adoption of new varieties, which could take more than 10 years[27] and restrict the availability of new varieties that are potentially adapted to future warmer conditions[28,29].

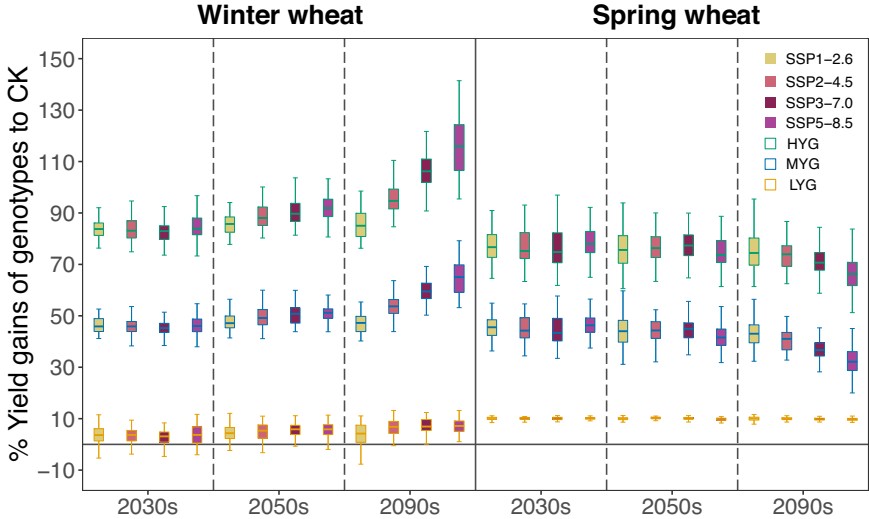

**Fig. 4 | Percentage changes in yield gains of four categories of genotype and CK in the same three future periods under four SSPs, projected by the model.** Boxplots show the median (the center line), 25th and 75th quartiles (lower and higher boxes), and smaller and largest value within 1.5 times interquartile range (lower and higher whiskers) in each period across six climate models. The filling color of boxplots indicates four SSPs. Within a column of an SSP, LYG is a yellow-bordered boxplot, MYG is a blue-bordered boxplot, and HYG is a red-bordered boxplot. Source data are provided as a Source Data file.

From a historical perspective, both mitigation and adaptation strategies are needed. First, mitigation toward SSP1-2.6 could be beneficial for maintaining breeding gains. The rainfed wheat-producing system in North America can be assured of a lower level of warming in SSP1-2.6 with the current breeding progress. Further warming in higher SSP scenarios would cause serious yield declines, with 3.6 °C for spring wheat and 6.0 °C warming for winter wheat. Reducing the climate warming rate is necessary to counteract climate stress, especially given that wheat production[30] and breeding gains[31] have stagnated in North America. Second, if we cannot limit the warming pace to the SSP1-2.6 scenario, a set of adaptation measures should be considered. Northward shifting of winter wheat production[32] and earlier planting of spring wheat[33] is helpful to reduce the negative effects of high-temperature extremes. Besides, we note that most breeding programs relied heavily on the phenotypic selection approach during the past decades represented by the nursery programs used for this analysis. This suggests that new breeding technologies, such as speed breeding[34], doubled-haploids[35], and genomic selection[36] are urgently needed and may have great value for accelerating the development of new varieties, particularly if major genetic determinants for the underlying process of heat stress adaptation are identified[37]. The exact value of these new breeding innovations as they pertain to heat tolerance should be clarified within the next decade. Continued assessment of new advanced lines should be evaluated in multiple environments to be exposed to local weather extremes which will opportunistically identify improved tolerance to temperature extremes.

We emphasize that our results do not account for the $CO_2$ fertilization effect[38,39]. This is because the relatively small importance of year-to-year $CO_2$ changes for yield variability results in highly uncertain estimates of $CO_2$ responses from such an effect[40]. Note, however, that our results on spring and wheat yield gains of variety adaptation quantified are quite robust regardless of $CO_2$ levels because our analysis provides pairs of data between advanced breeding lines and constant check varieties in each growing season. Furthermore, our model only represents the historical breeding progress and cannot reflect the potential accelerated breeding speed induced by new breeding technologies. More empirical and experimental quantifications of the benefits of new technologies are required in future investigations.

In summary, our study analyses yield responses of new wheat varieties to climate under rainfed conditions in North America using a multi-year multi-site dataset from breeding nurseries. Empirical results suggest that current genetic gain will not keep pace with future projected climate change, particularly for the spring wheat, and the future warming in high SSPs would offset yield gains with the continuation of current breeding progress. Slowing down the climate change rate and speeding up the variety of adaptations to future climate conditions are necessary. Integration of innovative technologies with traditional approaches in breeding for future climate can help to ensure the future productivity and climate resilience of wheat in a changing climate.

## Methods

### Wheat and climate data

Annual regional reports of the Southern Regional Performance Nursery[41], Northern Regional Performance Nursery[41], and Hard Red Spring Wheat Uniform Regional Nursery[42] in the years 1960–2018 were employed in the study. There are 62 sites for winter wheat and 30 sites for spring wheat, covering most wheat-producing areas in the Great Plains of North America (Fig. 1). In each site/season, new wheat breeding lines, as well as a check variety, are planted and grain yields are recorded after harvest for each entry and site/season. Each year the entries in the nurseries come from many geographically diverse breeding programs, but the constant check varieties were planted over 1960–2018. Kharkof is the check (CK) variety for winter wheat and Matquis for spring wheat. To represent the overall yield performances of new breeding lines in each season, we calculate the 50% percentile yield of the new experimental lines as the median-yielding genotype (MYG), 2.5% percentile as the low-yielding genotype (LYG), and 97.5% percentile as the high-yielding genotype (HYG).

The relationships between the yield trends of these genotypes and climate were analyzed using historical climate data. The daily minimum ($T_{min}$), maximum temperature ($T_{max}$), and precipitation (Prcp) data in these sites were obtained from the Daily Global Historical Climatology Network (GHCN-D)[43] and the Meteorological Service of Canada (MSC)[44] (Supplemental Tables 5–7). We calculated the cumulative exposures to growing-degree-day between the base and optimum growth temperature thresholds (GDD, °Cd, Eq. (1)), extreme-growing-degree days (EDD, °Cd, Eq. (2)) above the optimum growing temperature threshold, and freezing degree-day (FDD, °Cd, Eq. (3)) over growing season from hourly temperature ($T_h$) by fitting a cosine curve to daily $T_{min}$ and $T_{max}$ (Eq. (4)). To be comparable with climate and yield data, only the data pairs with both yield and climate were

included.

$$\text{GDD} = \sum_{h=1}^{N} \text{DD}_h; \text{DD}_h = \begin{cases} 0, & T_h < T_{\text{base}} \\ (T_h - T_{\text{base}})/24, & T_{\text{base}} \leq T_h \leq T_{\text{opt}} \\ (T_{\text{opt}} - T_{\text{base}})/24, & T_h > T_{\text{opt}} \end{cases} \quad (1)$$

$$\text{EDD} = \sum_{h=1}^{N} \text{DD}_h; \text{DD}_h = \begin{cases} 0, & T_h \leq T_{\text{opt}} \\ (T_h - T_{\text{opt}})/24, & T_h > T_{\text{opt}} \end{cases} \quad (2)$$

$$\text{FDD} = \sum_{h=1}^{N} \text{DD}_h; \text{DD}_h = \begin{cases} (T_{\text{frez}} - T_h)/24, & T_h \leq T_{\text{frez}} \\ 0, & T_h > T_{\text{frez}} \end{cases} \quad (3)$$

$$T_h = \frac{T_{\min} + T_{\max}}{2} + (T_{\max} - T_{\min})\frac{\cos(0.2618(h-14))}{2} \quad (4)$$

where $T_{\text{base}}$ and $T_{\text{opt}}$ are the base and optimum growing temperature thresholds specific to each growing phase of wheat (Supplementary Table 4), and $T_{\text{frez}}$ is the temperature causes freeze injury (Supplementary Table 4). Growing season accumulation of GDD, EDD, and FDD was calculated in each site-season pair. Unfortunately, only a small subset of sites recorded phenology data, and therefore trial-specific growing season lengths could not be used without omitting a large fraction of the data. The average plant and harvest dates in each site, therefore, were used based on Sacks et al.[45] for both winter and spring wheat. A fixed time window was often set for some temperature accumulation indices in climate impact studies[38,39], rather than the growing season of each individual year. This is because the latter would result in endogeneity in an analysis of FDD, GDD, and EDD on yields (e.g., warmer seasons may not have higher GDD than shorter growing seasons).

## Statistical yield models

Based on the historical data, we established statistical yield impact models for each genotype using the following fixed-effect regression model (Eq. (5)):

$$\log(Y_{i,t}) = \alpha_i + \alpha_{1i} \cdot t + \alpha_{2i} \cdot t^2 + \beta \cdot f(\text{Clim}_{i,t}) + \varepsilon_{i,t} \quad (5)$$

where $\log(Y_{i,t})$ is the natural logarithm of yield observation for the site $i$ in the year $t$, $\alpha_i$ is a site fixed effect accounting for time-invariant factors that vary across sites, $\alpha_{1i}$ is the site-specific linear time trend and $\alpha_{2i}$ is the site-specific quadratic time trend, which together account for changes over time that may influence yields, they are dummy variables to remove non-climate factors; $\beta$ is a vector of coefficients and $f(\text{Clim}_{i,t})$ is a vector of climate variables, $\varepsilon_{i,t}$ is the error term. In climate variables, we include the GDD, EDD, and FDD as temperature indices, and growing season accumulation of precipitation (Prcp) and its quadratic term as moisture index.

The regression coefficients determine the wheat yield responses as percentage changes in yields for each one-unit increase in climate variables. We used the bootstrapping method to determine the 95% confidence intervals by using a set of models generated from resampling (with replacement) the observations 1000 times. The 2.5th and 97.5th percentiles from the 1000 bootstrap replicates were selected as 95% confidence intervals for each regression coefficient. Potential multicollinearity among predictors was examined by variance inflation factors (Supplementary Fig. 7), which shows all these factors for temperature indices are <5 and do not present a serious multicollinearity problem to be cautious in the regression results[46].

## Future model projections

To project the future yield trend, we downloaded the climate projections from the Coupled Model Intercomparison Project Phase 6, with six climate models (ACCESS-ESM1-5, BCC-CSM2-MR, CNRM-CM6-1,

CNRM-ESM2-1, GFDL-ESM4, IPSL-CM6A-LR, Supplementary Table 8)[19] and four shared socioeconomic pathways (SSPs) (SSP1-2.6, SSP2-4.5, SSP3-7.0, and SSP5-8.5), and downscaled to each site to calculate climate variables following Eqs. (1)–(4).

We applied the statistical models developed for each category of genotypes by combining the climate projections. Future grain yield was projected in each year relative to the yield of the check variety over the 1960–2018 baseline period for winter and spring wheat under four SSPs. The projections for MYG, LYG, and HYG data account for the yield trends if we continue to breed wheat varieties at the historical pace, and the projection of CK yield represents the climate impact if no variety replacement occurred. The four SSPs represent greenhouse gas emission scenarios following degrees of climate policies. Besides, we also estimate the yield difference of MYG, LYG, and HYG relative to the CK varieties in each future time period to account for current variety replacement progress to future climate.

## Reporting summary

Further information on research design is available in the Nature Research Reporting Summary linked to this article.

## Data availability

The raw data are available on Zenodo repository: https://doi.org/10.5281/zenodo.7047191. Source data for each figure are provided with this paper. Source data are provided with this paper.

## Code availability

The codes are available on the Zenodo repository: https://doi.org/10.5281/zenodo.7018108.

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

## Acknowledgements
Tianyi Z. appreciate Qiao Kang and Zhi Chen for their supports in data preparation. Tianyi Z. is funded by the National Key Research and Development Project of China (2019YFA0607402) and the Youth Innovation Promotion Association of the Chinese Academy of Sciences (2018104). X. Yang is funded by the 2115 Talent Development Program of China Agricultural University.

## Author contributions
Tianyi Z. conceived the research idea; Tianyi Z., Y.H., R.P., D.G., X.D., Tao Z. collected and provided data; Tianyi Z., Y.H., R.D., Z.J., W.A., T.L. analyzed the data; X. Yue provided climate model projections; X. Yang provided some discussions; Tianyi Z. wrote the paper with contributions from all authors.

## Competing interests
The authors declare no competing interests.
