## [Peer Review File · Nature Communications]

Reviewers' Comments:

Reviewer #1:

Remarks to the Author:

General Comments

This paper provides a comprehensive analysis on the impact of climate change on winter and spring wheat yield using a large historic dataset and fixed-effect panel data regression models. Analysis using growing degree days (GDD), extreme degree days (EDD) and freezing degree days (FDD) are appropriate, and results provide valuable insights on the impact of these variables on wheat grain yield in current and future climates.

Weaknesses

The fixed-effect panel data regression model (Eq. 5) has a time (i.e. year) component, but the paper did not report any results on the site-specific time trend (linear and quadratic coefficients). The paper grouped the new experiment lines into three categories of high-yielding (HYG), median-yielding (MYG), and low-yielding (LYG) genotypes. But it was not clear which varieties were used in each year during 1960-2018 and it was not clear how yield gain through time (i.e year) was determined.

CO2 concentration was not included in the regression models and CO2 effect on yield was not included in the analysis and discussion.

Line 213-216 "The comparison of the yield of elite varieties with the constant CK variety in each year during the period 1960-2018 provides the first empirical assessment to benchmark the real-world breeding under historical climate change in the Great Plains of North America": It was not clear what were the breeding gain in yield over the years. Need to clarify how the HYG, MYG and LYG represent recent wheat releases as compared to the check variety.

Specific comments

Supplementary Table S4: What is the physiological basis for using different Tbase temperatures for different growth stages?

It was not clear whether the same check variety was used for winter wheat as well as spring wheat.

L30: by 1.9% less yield losses  with 1.9% less yield loss

L32: However, additional 0.5% yields were lost per 1oC warming for elite spring wheat  You mean on top of the 1.9% yield loss?

L82 Figure 1a: (000 ha)  (x1000 ha)

L82 Figure 1b and 1c: ton/ha you mean Metric ton /ha)?

L125: are slightly difference  are slightly different

L146-147: Finally, we apply the temperature thresholds of spring wheat in Supplementary Table S4 to winter wheat, and generate a very similar results  It was not clear how this was done (matching by stages?)

L164: for Kharkof  Not mentioned anywhere else in the paper

L165: Marquis  Not mentioned anywhere else in the paper

L225-226: The sentence is not clear

L229-231: Yield was estimated to decline by around 4.0-6.7% for winter wheat and 15.6-17.4% for spring wheat per each 100oCd increase in EDD  Yield is log-transformed before regression. Are you sure the regression coefficient represents proportional change?

L293: It is very difficult to mimic those rare new normal weather events in the traditional breeding approach  what you mean by 'rare new normal weather events' ?

L296: and more variety testing in multiple environments to prevent new varieties from leaving behind the future changes in climate  How multiple environments can address future changes in climate?

L372: as percentage changes in yield  Is this true with your log-transformed yield data in regression?

L374: resampling (with replacement)  what is the resampling size

L377: multicollinearity among predictors How about time (i.e. year) (Eq.5)?

Reviewer #2:

Remarks to the Author:

There is a lot of value to this study in terms of the impact of climate on wheat. I think there are some opportunities to explain the results better and provide a better "blueprint" for wheat breeders. Some of the approaches mentioned for breeders in the discussion section are outdated. My comments are below:

Line 64: Overall the authors do a poor job of providing a blueprint to "guide future breeding programs". Many of their suggestion in the discussion are outdated and oversimplify a complex breeding target.

Line 65: Use of 'severely' is an overstatement and not really true – plenty of studies in this area.

Overall there needs to be more intro on what impacts wheat yield, such as vernalization requirement in winter wheat vs spring wheat. Currently little to no information is provided to the readers.

Figure 1: I would like to see b and c as the trend for average performance across all the sites. Picking one and showing it can be very misleading and uninformative. How has the yield of the CK cultivar changed overtime? Previous studies have showed it increasing in yield. Showing % yield change instead of just yield would also be good so that it can be comparable to figure 3. Currently it is very hard to compare the two.

Line 118-120: So how much of the "less yield loss" can be attributed to each of these two components?

Line 124-126: Often times in these nurseries, locally adapted lines will be higher yielding. For example, Colorado breeding lines will perform better in Colorado compared to other breeding lines. I would imagine that this would be even more likely in winter wheat compared to spring wheat due to specific vernalization requirements for different sites. I think this idea of local adaptation could be influencing the results you are seeing here for winter wheat and the better performance to climate and compared to spring wheat. Also, higher yielding lines will often be more negatively impacted by heat stress compared to average or low yielding lines – even though they will still be higher yielding. Greater reduction vs. final yield. Do the authors have any thoughts or discussion on local adaptation?

Line 125: Grammar

Line 133-134: Can the authors speculate as to why this is happening? Is it changes in phenology such as heading date or maturity date?

Figure 3: I think the authors need to do a better job of projecting the next 80 years in the context of what has happened the last 60 years in their dataset. Though not shown (other than the Kansas example in figure 1) there has been a continuous increase in wheat yield from 1960 to 2020 and then based on this figure, we will see a rapid decline in this progress starting immediately. It's a little confusing and the lack of synergy between figure 1 and 3 does not help.

190-194: Why do the authors focus on MYG instead of HYG? Breeders do not release MYG. Also, there is a misuse of the word varieties. These are mostly breeding lines – some were released as varieties, but not the majority. Varieties are only those that are commercially available to a farmer.

225-228: Grammar issue here

Line 235-237: Breeding is not resulting in yield loss, climate is. This is misleading.

246-248: This has also been observed in CIMMYT Spring Wheat

286-287: With new technologies such as speed breeding, doubled-haploids, and genomic selection, variety development is done much more rapidly. These should be discussed if authors are making this claim.

Line 311-313: This is an oversimplification of the complexity of breeding – most methods now focus on a whole genome/genomic selection approaches, as marker assisted selection for a few genes will never be effective for a quantitative trait like heat tolerance.

341-344: What is the precedent for using these percentiles?

357-358: A lot of first person writing – i.e. We

358-360: Planting and harvest dates are likely available for most site years for this dataset and they can vary widely – why did the authors choose to use a previously published average? Seems like an oversimplification

Reviewer #3:

Remarks to the Author:

Thank you for preparing this interesting manuscript. I enjoyed reading it a lot. I agree that climate change impact modelling studies often point to the possibility of adapting the cultivars without discussing how effective such adaptation might be or whether the cultivars for that even exists yet. So I think your manuscript is an important contribution to that discussion.

I have mostly minor comments and feedback on specific sentences or how to present the results and some requests for more information, for instance about the yield data used. Please find them in the attached annotated manuscript.

I think that the discussion section could be shortened and improved. There are a lot of descriptions of results which should be removed or moved to only focus on the discussion of the results. I would avoid referring to specific figures and tables in this section. I appreciate that you want to keep the results section concise but it should not come at the cost of a longer discussion section. Perhaps work with sub-headings.

I am a bit concerned about the decision to use a published dataset for planting and harvest dates when you seem to have reported planting and harvest dates from the reports. There might be quite a deviation and you could not consider changes to these dates over time which could influence your results. Please explain your decision on this and potential bias introduced.

I am happy to review a revised version of the manuscript should it be required.

Good luck and kind regards,
Katharina Waha

Response letter to reviewers

We greatly appreciate these insightful comments and suggestions to our manuscript. We revised as you request and provided more analyses in the version, especially for CO₂ fertilization effect, potential uncertainties due to the lack of crop calendar data, and more discussions on new breeding technologies. Please check our point-by-point replies and associated revision in the new version.

All replies have been marked in blue, and our revision has been highlighted by yellow in the main text.

Reviewer #1 (Remarks to the Author):

General Comments

This paper provides a comprehensive analysis on the impact of climate change on winter and spring wheat yield using a large historic dataset and fixed-effect panel data regression models. Analysis using growing degree days (GDD), extreme degree days (EDD) and freezing degree days (FDD) are appropriate, and results provide valuable insights on the impact of these variables on wheat grain yield in current and future climates.

Reply: We greatly appreciate your suggestions. We have revised based on your comments, and provided an analysis for CO₂ fertilization effect. Please check our point-by-point reply:

Weaknesses

The fixed-effect panel data regression model (Eq. 5) has a time (i.e. year) component, but the paper did not report any results on the site-specific time trend (linear and quadratic coefficients).

Reply: In panel data regression model, the values of site, site-year and site-year² are "dummy variables to remove non-climate factors" (lines 360). We attached a XLSX file to show reviewers the full regression coefficients. Please check the values in Full_RegCoef.xlsx.

In our manuscript, we focused on how changes in climate affect yields. Therefore, we only reported the regression coefficients of climate variables in the main text.

The paper grouped the new experiment lines into three categories of high-yielding (HYG), median-yielding (MYG), and low-yielding (LYG) genotypes. But it was not clear which varieties were used in each year during 1960-2018 and it was not clear how yield gain through time (i.e. year) was determined.

Reply: Sorry about the confusion. The advanced breeding lines entered into the nursery programs studied each year are different than the previous year. Therefore, we did not fix a certain breeding line. Instead, we used a relative term to quantify various yield levels of HYG, MYG and LYG. More specifically, HYG is the 97.5% percentile grain yield for advanced breeding lines each year, MYG is 50% percentile, and LYG is 2.5% percentile. To address this concern, we added the following sentence: "On the other hand, the advanced breeding lines entered each year were different from the previous year, which reflects the ongoing nature of wheat breeding itself over the study period." in lines 82-84.

If the temperature stress can be completely canceled out by variety replacement, yield sensitivity of advanced breeding lines to climate should be lower than results based on CK yield data series, and

vice versa. We added the following sentence: " Therefore, the difference in yield sensitivity (advanced breeding lines vs. check variety) can be defined as effectiveness of the current variety development under climate change, providing a footprint of the actual wheat breeding effort." (lines 85-87)

CO₂ concentration was not included in the regression models and CO₂ effect on yield was not included in the analysis and discussion.

Reply: We appreciate the reviewer bringing up this point and agree that it is important to address. Earlier study (Lobell and Field, 2007) has found that CO₂ effect is difficult to quantify based on a statistical approach using historical crop statistics because the relatively small importance of year-to-year CO₂ changes for yield variability results in the high uncertain estimates of CO₂ responses from such approach.

To demonstrate this, we followed the approach of Lobell and Field (2007) and included CO₂ item in a panel data regression model (the Table R1 in next page). Similar to Lobell and Field (2007), for the US we find that:

(1) The regression coefficient of CO₂ is negative and insignificant in many cases, which is physiologically irrational with the CO₂ fertilization effects. Our estimate is the same with Fig. 6 in Lobell and Field (2007) paper, in which they also found negative regression in US wheat. We believe this is because the year-to-year CO₂ variability is much smaller than climate impact, and very difficult to have clear effects detected based on statistical model.

(2) The regression coefficients of other climate variables are not affected by inclusion of CO₂, which reflects robustness of our estimation for other climate variables.

The above points indicate that estimating a CO₂ fertilization effect directly in a regression framework is unlikely to be accurate. As an alternative approach, we instead take advantage of the study design of comparing HYG, MYG, and LYG yields to a check variety. We have rewritten our results to present our findings as the difference between pairs of data accordingly. Presented in this way, the results on spring and wheat yield gains of variety adaptation quantified is robust regardless of CO₂ levels because CO₂ levels within a year are constant across varieties within each growing season.

We added the sentences: " Finally, we note that studies based on panel data model do not often consider CO₂ fertilizations^{44,45}. This is because the relatively small importance of year-to-year CO₂ changes for yield variability results in the highly uncertain estimates of CO₂ responses from such approach⁴⁷. Additionally, results on spring and wheat yield gains of variety adaptation quantified is robust regardless of CO₂ levels because our analysis provides pairs of data between advanced breeding lines and constant check varieties in each growing season. We encourage future field investigations and variety testing on different CO₂ levels." in lines 373-380.

Lobell, D., Field, C., Estimation of the carbon dioxide (CO₂) fertilization effect using growth rate anomalies of CO₂ and crop yields since 1961. *Glob. Change Biol.* **14**, 39-45 (2008).

Table R1. Regression coefficients of winter and spring wheat without CO₂ (the results shown in our manuscript) and with CO₂ (the results we add CO₂ item in regression model). The number in parenthesis is the standard error in linear regression analysis. $p < 0.05^*$, $p < 0.01^{**}$

Variables		Winter wheat				Spring wheat			
		CK	MYG	HYG	LYG	CK	MYG	HYG	LYG
Without CO ₂	FDD	-3.80e-04*	-6.4e-04**	-4.6e-04**	-1.05e-03**	-2.42e-02**	-1.80e-02**	-1.86e-02**	-1.86e-02**
		(1.69e-04)	(1.46e-04)	(1.23e-04)	(2.13e-04)	(7.86e-03)	(6.15e-03)	(5.67e-03)	(7.87e-03)
	GDD	-5.22e-05	-7.85e-05	-1.0e-05	3.20e-05	4.80e-05	1.46e-04	1.81e-04	1.69e-05
		(1.42e-04)	(1.23e-04)	(1.0 e-04)	(1.79e-04)	(2.62e-04)	(2.05e-04)	(1.89e-04)	(2.62e-04)
	EDD	-6.3e-04**	-4.7e-04**	-4.0e-04**	-6.7e-04**	-1.57e-03**	-1.74e-03**	-1.67e-03**	-1.56e-03**
		(1.70e-04)	(1.47e-04)	(1.2e-04)	(2.15e-04)	(3.20e-04)	(2.51e-04)	(2.31e-04)	(3.20e-04)
	Prcp	2.26e-03**	2.79e-03**	2.48e-03**	2.91e-03**	2.42e-03**	2.87e-03**	2.78e-03**	3.03e-03**
	(3.34e-04)	(2.87e-04)	(2.35e-04)	(4.21e-04)	(6.13e-04)	(4.81e-04)	(4.43e-04)	(6.14e-04)	
Prcp ²		-2.52e-06**	-2.61e-06**	-2.21e-06**	-2.88e-06**	-5.50e-06**	-5.37e-06**	-4.86e-06**	-6.58e-06**
		(3.33e-07)	(2.87e-07)	(2.35e-07)	(4.19e-07)	(9.90e-07)	(7.75e-07)	(7.15e-07)	(9.91e-07)
With CO ₂	CO ₂	-0.035*	-0.016	-0.0084	-0.046*	-0.026	-0.0168	-0.0172	-0.035
		(0.017)	(0.014)	(0.012)	(0.021)	(0.020)	(0.0158)	(0.0146)	(0.02)
	FDD	-3.74e-04*	-6.4e-04**	-4.6e-04**	-1.05e-03**	-2.39e-02**	-1.78e-02**	-1.85e-02**	-1.84e-02**
		(1.69e-04)	(1.46e-04)	(1.23e-04)	(2.13e-04)	(7.85e-03)	(6.15e-03)	(5.67e-03)	(7.86e-03)
	GDD	-1.64e-05	-6.18e-05	-9.49e-05	7.84e-05	5.26e-05	1.46e-04	1.84e-04	2.31e-05
		(1.43e-04)	(1.23e-04)	(1.0 e-04)	(1.8e-04)	(2.62e-04)	(2.05e-04)	(1.89e-04)	(2.62e-04)
	EDD	-6.28e-04**	-4.7e-04**	-3.9e-04**	-6.8e-04**	-1.5e-03**	-1.69e-03**	-1.63e-03**	-1.47e-03**
	(1.70e-04)	(1.47e-04)	(1.2e-04)	(2.14e-04)	(3.24e-04)	(2.54e-04)	(2.34e-04)	(3.24e-04)	
Prcp		2.22e-03**	2.79e-03**	2.48e-03**	2.89e-03**	2.42e-03**	2.87e-03**	2.78e-03**	3.03e-03**
		(3.33e-04)	(2.88e-04)	(2.35e-04)	(4.2e-04)	(6.13e-04)	(4.81e-04)	(4.43e-04)	(6.14e-04)
Prcp ²		-2.51e-06**	-2.61e-06**	-2.21e-06**	-2.87e-06**	-5.48e-06**	-5.36e-06**	-4.86e-06**	-6.57e-06**
		(3.33e-07)	(2.87e-07)	(2.35e-07)	(4.19e-07)	(9.90e-07)	(7.75e-07)	(7.15e-07)	(9.90e-07)

Line 213-216 “The comparison of the yield of elite varieties with the constant CK variety in each year during the period 1960-2018 provides the first empirical assessment to benchmark the real-world breeding under historical climate change in the Great Plains of North America”: It was not clear what were the breeding gain in yield over the years. Need to clarify how the HYG, MYG and LYG represent recent wheat releases as compared to the check variety.

Reply: Sorry about the confusion. The purpose of the work is to compare yield sensitivity of advanced breeding lines to a constant check variety, which is defined as effectiveness of the current variety development under climate change. Therefore, we rewrote the sentence which reads "We provide the first empirical assessment to benchmark the real-world breeding effect based on long-term breeding nursery observations from multiple sites in which the climate exposures of advanced breeding-lines and constant check varieties can be strictly controlled in the Great Plains of North America." in lines 244-247.

To show how much the yield sensitivity of our breeding nurseries can represent the actual sensitivity for farmer's yields (that is, the real wheat releases), we compare the yield sensitivity to 1°C warming based on our data with the results based on county-level yield statistics. We rewrote the sentence: "when the above statistical analysis was repeated but used the county-level yield statistics, the yield responses of MYG to 1°C warming derived here were close to the estimation from county-level yield statistics (Supplementary Fig. S2). This reflected that the response of MYG can approximate and represent the yield sensitivity in farmer's fields at regional scale. " in lines 151-155.

Supplementary Figure S2. Comparison of model estimates of yield response to 1°C warming using regression models for data of nurseries and county-level yield statistics from National Agricultural Statistics Service and Statistics Canada. The error bar indicates the 95% CI derived from bootstrap analysis.

Specific comments

Supplementary Table S4: What is the physiological basis for using different Tbase temperatures for different growth stages?

Reply: Earlier studies showed that the critical temperature thresholds vary with growth phases of wheat. Please find these references in the caption of supplementary Table S4: "The base and optimum growing temperature in each phenological phases for winter and spring wheat. The threshold is based on Narciso et al.¹¹; Shroyer et al.¹², Porter and Gawith¹³, Tack et al.⁵ and Saiyed et al.¹⁴"

Tack, J., Barkley, A., Nalley, L., Effect of warming temperatures on US wheat yields. Proc. Natl. Acad.

- Sci. U.S.A. 112, 6931-6936. (2015).
- Narciso, G., Ragni, P., Venturi, A., Agrometeorological Aspects of crops in Italy, Spain and Greece. Joint Research Centre, Commission of the European Communities, Brussels–Luxembourg (1992).
- Shroyer, J., Mikesell, M., Paulsen, G., Spring freeze injury to Kansas wheat. Agricultural Experiment Station and Cooperative Extension Service (1995).
- Porter, J., Gawith, M., Temperatures and the growth and development of wheat: a review. Euro. J. of Agron., 10, 23-36 (1999).
- Saiyed, M., Bullock, R., Sapirstein, D., Finlay, J., Jarvis, K. Thermal time models for estimating wheat phenological development and weather-based relationships to wheat quality. Can. J. Plant Sci. 89, 429-439 (2009).

It was not clear whether the same check variety was used for winter wheat as well as spring wheat.

Reply: Sorry about the confusion. To address this, we rewrote the sentence: " The variety Kharkof was used as the long-term winter wheat check, and the variety Marquis as the long-term spring wheat check. The two check varieties were planted in each site throughout the study period, which could be viewed as yield variability without variety replacement. " in lines 79-82 and lines 324-325.

L30: by 1.9% less yield losses  with 1.9% less yield loss

Reply: As request by reviewer#3, the sentence has been removed. We now show yield decline for advanced breeding lines and check variety directly: "Results shows that yields were declined by 3.6%/°C for elite winter wheat breeding lines, compared with -5.4%/°C when variety is held constant, reflecting a superior climate-resilience. However, advanced spring wheat breeding lines demonstrate 7.5% yield reduction per 1°C warming. That is more sensitive than the value of -7.1%/°C with constant variety planted, indicating an undermined climate-resilience for spring wheat.". Please check lines 32-37.

L32: However, additional 0.5% yields were lost per 1oC warming for elite spring wheat  You mean on top of the 1.9% yield loss?

Reply: The "1.9% less yield losses" is for winter wheat and "0.5% yield" is for spring wheat. However, as request by reviewer#3, the sentence has been removed. We now show yield decline for advanced breeding lines and check variety directly: "Results shows that yields were declined by 3.6%/°C for elite winter wheat breeding lines, compared with -5.4%/°C when variety is held constant, reflecting a superior climate-resilience. However, advanced spring wheat breeding lines demonstrate 7.5% yield reduction per 1°C warming. That is more sensitive than the value of -7.1%/°C with constant variety planted, indicating an undermined climate-resilience for spring wheat.". Please check lines 32-37.

L82 Figure 1a: (000 ha)  (□1000 ha)

Reply: Revised as request. Please check Fig. 1. Based on requirement of Nature, we changed it to "1,000 ha".

L82 Figure 1b and 1c: ton/ha you mean Metric ton /ha)?

Reply: Yes, it is metric tonnes per ha. We have specified the unit (mt/ha) in the figure. Please check Supplementary Fig. S1.

L125: are slightly difference  are slightly different

Reply: Revised as request. Please check line 127.

L146-147: Finally, we apply the temperature thresholds of spring wheat in Supplementary Table S4 to winter wheat, and generate a very similar results  It was not clear how this was done (matching by stages?)

Reply: Sorry about the confusion. As you said, we match growth phases for heading to maturity. Before that stage, we believe some genetic difference still exists as their genetic basis is quite different between winter and spring wheat (i.e. winter wheat needs to experience winter but spring wheat does not). To address the confusion, we rewrote the sentence: "Finally, we apply the temperature thresholds of spring wheat in Supplementary Table S4 to winter wheat from heading to maturity, and generate a very similar results for winter wheat yield response to 1°C warming" (lines 161-163).

L164: for Kharkof  Not mentioned anywhere else in the paper

Reply: Sorry about the confusion. To address this, we rewrote the sentence: " The variety Kharkof was used as the long-term winter wheat check, and the variety Marquis as the long-term spring wheat check. The two check varieties were planted in each site throughout the study period, which could be viewed as yield variability without variety replacement. " in lines 79-82 and lines 324-325.

L165: Marquis  Not mentioned anywhere else in the paper

Reply: Sorry about the confusion. To address this, we rewrote the sentence: " The variety Kharkof was used as the long-term winter wheat check, and the variety Marquis as the long-term spring wheat check. The two check varieties were planted in each site throughout the study period, which could be viewed as yield variability without variety replacement. " in lines 79-82 and lines 324-325.

L225-226: The sentence is not clear

Reply: Sorry about the confusion. We rewrote the sentence: " Warming poses a more harmful effect on yield of spring wheat breeding, resulting primarily from greater sensitivity to EDD of the advanced spring wheat breeding lines than CK." in lines 251-253.

L229-231: Yield was estimated to decline by around 4.0-6.7% for winter wheat and 15.6-17.4% for spring wheat per each 100oCd increase in EDD  Yield is log-transformed before regression. Are you sure the regression coefficient represents proportional change?

Reply: Yes. The regression coefficients represent the percentage change in dependent variables for every one-unit increase in the independent variable. Please check the trivial example below:

Suppose the fitted model is

$$\log(\text{weight}) = 2.14 + 0.00055 \text{ height}$$

The estimated coefficient for height is 0.00055, so we would say that an increase of one unit in height is associated with a $100 \times (e^{0.00055} - 1) \approx 0.055$ percent change in weight.

To address this concern, we added the following sentence: "The regression coefficients determine the wheat yield response as percentage changes in yields for each one-unit increase in climate variables." in lines 364-365.

L293: It is very difficult to mimic those rare new normal weather events in the traditional breeding approach  what you mean by 'rare new normal weather events'?

Reply: In breeding programs, breeders still rely on phenotypic selection approach. In the actual field conditions, it is very difficult to create an environment with very high temperature (i.e. the "rare new normal weather events"). Therefore, it will be very difficult for breeders to select the breeding lines with potential heat tolerance using such a traditional breeding approach.

As request by reviewer#2, we have rewritten the part and discuss more state-of-the-art breeding technologies, which reads" we note that most breeding programs relied heavily on phenotypic selection approach during the past decades represented by the nursery programs used for this analysis. This suggests that new breeding technologies are required to be implement now, such as speed breeding³⁵, doubled-haploids³⁶, and genomic selection³⁷ may have great value for accelerating the development of new varieties, particularly if major genetic determinants for the underlying process of heat stress adaptation are identified³⁸. The exact value of these new breeding innovations as they pertain to heat tolerance should be clarified within the next decade. Continued assessment of new advanced lines should be evaluated in multiple environments to be exposed to local weather extremes which will opportunistically identify improved tolerance to temperature extremes." in lines 289-299.

L296: and more variety testing in multiple environments to prevent new varieties from leaving behind the future changes in climate  How multiple environments can address future changes in climate?

Reply: To us, we need to test the advanced breeding lines performance under various heat stress environments vs. normal temperature regime (i.e. the "multiple environments"), and identify the major genetic determinants for the underlying process of heat stress.

As request by reviewer#2, we have rewritten the part and discuss more state-of-the-art breeding technologies, which reads" we note that most breeding programs relied heavily on phenotypic selection approach during the past decades represented by the nursery programs used for this analysis. This suggests that new breeding technologies are required to be implement now, such as speed breeding³⁵, doubled-haploids³⁶, and genomic selection³⁷ may have great value for accelerating the development of new varieties, particularly if major genetic determinants for the underlying process of heat stress adaptation are identified³⁸. The exact value of these new breeding innovations as they pertain to heat tolerance should be clarified within the next decade. Continued assessment of new advanced lines should be evaluated in multiple environments to be exposed to local weather extremes which will opportunistically identify improved tolerance to temperature extremes." in lines 289-299.

L372: as percentage changes in yield  Is this true with your log-transformed yield data in regression?

Reply: Yes. The regression coefficients represent the percentage change in dependent variables for every one-unit increase in the independent variable. Please check the trivial example below:

Suppose the fitted model is

$$\log(\text{weight}) = 2.14 + 0.00055 \text{ height}$$

The estimated coefficient for height is 0.00055, so we would say that an increase of one unit in height is associated with a $100 \times (e^{0.00055} - 1) \approx 0.055$ percent change in weight.

To address this concern, we added the following sentence: "The regression coefficients determine the wheat yield response as percentage changes in yields for each one-unit increase in

climate variables.” in lines 364-365.

L374: resampling (with replacement)  what is the resampling size

Reply: This is the bootstrapping method that often used to determine the uncertainty of regression model in climate impact studies (Lobell et al., 2012). Bootstrapping is a method of resample. The idea is to use the observed sample to estimate the population distribution. Then samples can be drawn from the estimated population and the sampling distribution of any type of estimator can itself be estimated.

For example, the observed sample is $\begin{pmatrix} 1 & 1.1 \\ 2 & 2.2 \\ 3 & 2.9 \\ 4 & 3.5 \end{pmatrix}$. The first column is the independent data series

and the second column is the dependent data series. We have a regression equation, $y=0.79x+0.45$.

Then we draw randomly by resampling with replacement. The first resample might be $\begin{pmatrix} 3 & 2.9 \\ 2 & 2.2 \\ 3 & 2.9 \\ 1 & 1.1 \end{pmatrix}$, and

the second might be $\begin{pmatrix} 4 & 3.5 \\ 2 & 2.2 \\ 2 & 2.2 \\ 1 & 1.1 \end{pmatrix}$We can repeat these by 1000 times and for each time we have a new

regression equation. The 1000 sets of regression coefficients can represent the uncertainty of regression model due to resampling size.

Lobell, D., Sibley, A., Ortiz-Monasterio, I., Extreme heat effects on wheat senescence in India. Nat. Clim. Change. 2, 186-189 (2012).

L377: multicollinearity among predictors How about time (i.e. year) (Eq.5)?

Reply: We add both Year and Year² in model, so they have a strong multicollinearity by themselves for sure. However, we did not project model by the Year and Year² items but only use temperature-related climate indices whose multicollinearity problem is weak based on VIF.

Reviewer #2 (Remarks to the Author):

There is a lot of value to this study in terms of the impact of climate on wheat. I think there are some opportunities to explain the results better and provide a better "blueprint" for wheat breeders. Some of the approaches mentioned for breeders in the discussion section are outdated. My comments are below:

Reply: We greatly appreciate your positive comments. We have revised, and provides more discussions on the new breeding technologies and how to develop heat tolerance properties in an effective manner. Please check our point-by-point reply.

Line 64: Overall the authors do a poor job of providing a blueprint to "guide future breeding programs". Many of their suggestion in the discussion are outdated and oversimplify a complex breeding target.

Reply: Based on this suggestion, we have rewritten the part and discuss more state-of-the-art breeding technologies, which reads "we note that most breeding programs relied heavily on phenotypic selection approach during the past decades represented by the nursery programs used for this analysis. This suggests that new breeding technologies are required to be implement now, such as speed breeding, doubled-haploids, and genomic selection may have great value for accelerating the development of new varieties, particularly if major genetic determinants for the underlying process of heat stress adaptation are identified. The exact value of these new breeding innovations as they pertain to heat tolerance should be clarified within the next decade. Continued assessment of new advanced lines should be evaluated in multiple environments to be exposed to local weather extremes which will opportunistically identify improved tolerance to temperature extremes." in lines 289-299.

Line 65: Use of 'severely' is an overstatement and not really true – plenty of studies in this area.

Reply: Earlier studies focus on the comparison between yield itself of new and check varieties. Here, we focus on the yield sensitivities to climate with and without genetic advancement, which has not been studied much based on our knowledge.

We revised as request: "To date, the relevant understanding remains lacking because of the scarcity of long-term field observations enabling comparison between yield sensitivities to climate with and without genetic advancement." in lines 68-71.

Overall there needs to be more intro on what impacts wheat yield, such as vernalization requirement in winter wheat vs spring wheat. Currently little to no information is provided to the readers.

Reply: Thanks for this suggestion. We introduced several important references to processes of warming that reduces wheat yields: "Many plant species are negatively affected by high temperature extremes, especially during floral stage¹¹, and often have a shortened grain filling period¹², less biomass accumulation¹⁰ and consequently lower grain yields^{3,4,5,6}" in lines of 55-58.

Besides, we also stated that, to understand the different response of winter and spring wheat, "Such differences in responses show the need to understand further the underlying genetic process involved, such as specific vernalization requirements and variety responses to heat/cold stress between winter wheat versus spring wheat. More control field experiments are needed to examine the genetic basis across winter and spring wheat genotypes in future investigations." in lines of 262-266.

Figure 1: I would like to see b and c as the trend for average performance across all the sites. Picking one and showing it can be very misleading and uninformative. How has the yield of the CK cultivar

changed overtime? Previous studies have showed it increasing in yield. Showing % yield change instead of just yield would also be good so that it can be comparable to figure 3. Currently it is very hard to compare the two.

Reply: We agree with the concern. Indeed, many earlier studies have found increase in CK yield. From our data, the yield trend of CK variety is also increasing for winter and spring wheat over time (Figure 1R below). However, we should note that the site number is not constant in each year. For example, for SRPN, there were 15 sites in 1960 but the number of sites is 24 in 2018. And some sites were not reported in some years and some new site was included in the breeding program. Therefore, the time trend of the average yield across all the sites not only include climate signal but also includes some information of changing site. So we believe this time trend for average yield cannot reflect yield response to climate.

In our Fig. 2, the regression coefficients are estimated based on all data, so this is the demonstration of climate impact on yields.

To address this concern, we picked two sites just for demonstrating how we determine CK, LYG, MYG and HYG yield series to readers. Besides, we have moved them to Supplementary Fig. S1 to avoid such misleading.

Fig. R1. The average yield of check varieties in each year across all sites for winter wheat (a) and spring wheat (b).

Line 118-120: So how much of the “less yield loss” can be attributed to each of these two components?

Reply: Sorry about the confusion. We rewrote the sentence " For example, with 1°C warming, yield response is lowered by 1.6% to the associated increase in EDD for advanced breeding lines relative to check variety (MYG: -4.5% vs. CK: -6.1%). One-degree warming brings 0.71% greater yield benefits due to the associated decrease in FDD for MYG relative to CK (MYG: +1.58% vs. CK: +0.87%)." in lines 122-126.

Line 124-126: Often times in these nurseries, locally adapted lines will be higher yielding. For example, Colorado breeding lines will perform better in Colorado compared to other breeding lines. I would imagine that this would be even more likely in winter wheat compared to spring wheat due to specific vernalization requirements for different sites. I think this idea of local adaptation could be influencing the results you are seeing here for winter wheat and the better performance to climate and compared to spring wheat. Also, higher yielding lines will often be more negatively impacted by heat stress compared to average or low yielding lines – even though they will still be higher yielding. Greater reduction vs. final yield. Do the authors have any thoughts or discussion on local adaptation?

Reply: We agree that advanced breeding lines for a particular location will be better adapted there, i.e. the local-adaptation, but they represent only a fraction of the entries in any given year. The entries in

the nurseries come from many geographically diverse breeding programs. Therefore, the advanced breeding lines tested in experiments has avoid local-adaptation issue. We have added the sentence " Each year the entries in the nurseries come from many geographically diverse breeding programs (The source of the breeding lines was listed as entries in each report of the NRPN, SRPN and HRSWURN)". in lines 321-322.

In our results, it is not always the case for higher yielding genotypes to suffer more from heat injury than lower yielding genotypes. For example, the yield decline of MYG is less than CK for winter wheat, and the MYG is the higher yielding genotypes relative to the CK variety (Fig. 2).

Line 125: Grammar

Reply: We changed to "different". Please check line 127.

Line 133-134: Can the authors speculate as to why this is happening? Is it changes in phenology such as heading date or maturity date?

Reply: Based on the data availability of USDA, only a small subset of sites recorded sowing and harvest dates. So, we cannot show how wheat phenology change over 1960-2018.

Our analysis focuses on the difference of yield sensitivities of climate on different genotypes, but attribution of reasons is not possible using our current yield data only. To understand the different response of winter and spring wheat, "Such differences in responses show the need to understand further the underlying genetic process involved, such as specific vernalization requirements and variety responses to heat/cold stress between winter wheat versus spring wheat. More control field experiments are needed to examine the genetic basis across winter and spring wheat genotypes in future investigations." (lines of 262-266).

Figure 3: I think the authors need to do a better job of projecting the next 80 years in the context of what has happened the last 60 years in their dataset. Though not shown (other than the Kansas example in figure 1) there has been a continuous increase in wheat yield from 1960 to 2020 and then based on this figure, we will see a rapid decline in this progress starting immediately. It's a little confusing and the lack of synergy between figure 1 and 3 does not help.

Reply: Sorry for the confusion. The yield change we project in Fig. 3 is not actual yield. It is the yield changes due to climate change under four SSPs. Actual yield is not predicable because the non-climate factors (changes in agronomy technologies, like fertilization and irrigation etc.) is not possible to predict.

To address the confusion, we rewrote the figure captions "Figure 3. Model projection of wheat yield changes due to changes in climate of four categories of genotype under four SSPs, relative to the yield of CK at the baseline period." in lines 197-199.

190-194: Why do the authors focus on MYG instead of HYG? Breeders do not release MYG. Also, there is a misuse of the word varieties. These are mostly breeding lines – some were released as varieties, but not the majority. Varieties are only those that are commercially available to a farmer.

Reply: Based on your suggestion, we revised Fig. 4 and put LYG, MYG and HYG together.

We also compared the yield sensitivity of CK, LYG, MYG and HYG with county-level yield statistics. We found the "yield responses of MYG to 1°C warming derived here were close to the estimation from county-level yield statistics (Supplemental Fig. S2)" (lines 151-155). We checked the

whole manuscript to change varieties to breeding lines, as request.

225-228: Grammar issue here

Reply: Revise as request: " Warming poses a more harmful effect on yield of spring wheat breeding, resulting primarily from greater sensitivity to EDD of the advanced spring wheat breeding lines than CK." in lines of 251-253.

Line 235-237: Breeding is not resulting in yield loss, climate is. This is misleading.

Reply: Thanks for pointing out this. We rewrote: "Spring wheat is projected to suffer more yield losses compared to winter wheat with future temperature increases" in lines 258-260.

246-248: This has also been observed in CIMMYT Spring Wheat

Reply: Yes. "A recent global-scale study²⁷ based on performance of CIMMYT spring wheat and durum experimental lines also pointed out that climate change may impede the rate of genetic gains in both spring wheat and durum wheat breeding" (lines 272-274).

286-287: With new technologies such as speed breeding, doubled-haploids, and genomic selection, variety development is done much more rapidly. These should be discusses if authors are making this claim.

Reply: Appreciate your suggestion. Based on this suggestion, we have rewritten the part and discuss more state-of-the-art breeding technologies, which reads" we note that most breeding programs relied heavily on phenotypic selection approach during the past decades represented by the nursery programs used for this analysis. This suggests that new breeding technologies are required to be implement now, such as speed breeding³⁵, doubled-haploids³⁶, and genomic selection³⁷ may have great value for accelerating the development of new varieties, particularly if major genetic determinants for the underlying process of heat stress adaptation are identified³⁸. The exact value of these new breeding innovations as they pertain to heat tolerance should be clarified within the next decade. Continued assessment of new advanced lines should be evaluated in multiple environments to be exposed to local weather extremes which will opportunistically identify improved tolerance to temperature extremes." in lines 289-299.

Line 311-313: This is an oversimplification of the complexity of breeding – most methods now focus on a whole genome/genomic selection approaches, as marker assisted selection for a few genes will never be effective for a quantitative trait like heat tolerance.

Reply: That is a very interesting point. To our knowledge, the reason that traditional genome selection is ineffective for heat tolerance breeding might reflect these traditional genome selection algorithms do not consider the underlying processes that environmental factors affect plant growth and development, and as such, have limited ability to capture Genotype × Environment interactions. Furthermore, robust phenotypic data for heat tolerance is required to validate a training set which continues to be difficult to generate.

Recently, Dr. Tao Li, one of our co-authors, is working on integration of genomics with process-based crop modeling to overcome the disadvantages. The new technology is still under testing, for example, to predict phenology traits of rice varieties under different temperature regimes (Yang et al., 2022). However, this breeding technology is too young to implement right now. The exact value of

these new breeding innovations requires to be evaluating and could be clarified within the next decade.

We have rewritten the part and discuss more state-of-the-art breeding technologies, which reads "we note that most breeding programs relied heavily on phenotypic selection approach during the past decades represented by the nursery programs used for this analysis. This suggests that new breeding technologies are required to be implement now, such as speed breeding³⁵, doubled-haploids³⁶, and genomic selection³⁷ may have great value for accelerating the development of new varieties, particularly if major genetic determinants for the underlying process of heat stress adaptation are identified³⁸. The exact value of these new breeding innovations as they pertain to heat tolerance should be clarified within the next decade. Continued assessment of new advanced lines should be evaluated in multiple environments to be exposed to local weather extremes which will opportunistically identify improved tolerance to temperature extremes." in lines 289-299.

Yang, Y., Wilson, L., Li, T. *et al.*, Integration of genomics with crop modeling for predicting rice days to flowering: A multi-model analysis. *Field Crop Res.* 276, 108394 (2022).

341-344: What is the precedent for using these percentiles?

Reply: Earlier studies often use the yield of five most productive breeding lines. But, here, we prefer a relative term to define yield levels. This is because the number of breeding lines to test is very different in each year. For example, there are only 7 entries in Colby in 1962 but the number is 50 in 2018. Therefore, we set the HYG as the 97.5% percentile of yields for advanced breeding lines and LYG as the 2.5% percentile. So, 95% entries could be included.

357-358: A lot of first person writing – i.e. We

Reply: We have revised: " Growing season accumulation of GDD, EDD and FDD were calculated in each site-season pair." in lines 343-344.

358-360: Planting and harvest dates are likely available for most site years for this dataset and they can vary widely – why did the authors choose to use a previously published average? Seems like an oversimplification

Reply: We agree with the concern. However, based on the data availability of USDA, only a small subset of sites recorded sowing and harvest dates. .

We have revised: "Unfortunately, only a small subset of sites recorded phenology data, and therefore trial-specific growing season lengths could not be used without omitting a large fraction of the data. The average plant and harvest dates in each site therefore were used based on Sacks et al. for both winter and spring wheat. A fixed time window was often set for some temperature accumulation indices in climate impact studies, rather than the growing season of each individual year. This is because the latter would result in endogeneity in an analysis of FDD, GDD and EDD on yields (e.g. warmer season may not have higher value of GDD as shorter growing season). " in lines 345-352.

Despite the endogeneity problem, we can still test whether this will influence our results. We randomly selected the planting and harvest dates in different years for a certain site based on the calendar range reported in Sacks et al. And we repeat the process for 1000 times, and did the regression analysis (Fig. R2). We found that:

(1) The yield impact by 1°C warming is slightly smaller than the results in Fig. 2. This reflects the endogeneity problem, and the degree-day cannot well reflect the actual temperature difference

between years.

(2) The major conclusion in the main text still holds: the yield decline of MYG is smaller than CK for winter wheat, the results is contrary for spring wheat. Therefore, we believe crop calendar does not influence our major conclusion.

Fig. R2. Comparison of model estimates of yield response to 1°C warming using regression models based on the sites with randomly selected planting and harvest dates.

Reviewer #3 (Remarks to the Author):

Thank you for preparing this interesting manuscript. I enjoyed reading it a lot. I agree that climate change impact modelling studies often point to the possibility of adapting the cultivars without discussing how effective such adaptation might be or whether the cultivars for that even exists yet. So I think your manuscript is an important contribution to that discussion.

Reply: Great appreciation from us. We have moved some part of paragraphs to result part and make discussion shorter. Please check our point-by-point reply.

I have mostly minor comments and feedback on specific sentences or how to present the results and some requests for more information, for instance about the yield data used. Please find them in the attached annotated manuscript.

I think that the discussion section could be shortened and improved. There are a lot of descriptions of results which should be removed or moved to only focus on the discussion of the results. I would avoid referring to specific figures and tables in this section. I appreciate that you want to keep the results section concise but it should not come at the cost of a longer discussion section. Perhaps work with sub-headings.

I am a bit concerned about the decision to use a published dataset for planting and harvest dates when you seem to have reported planting and harvest dates from the reports. There might be quite a deviation and you could not consider changes to these dates over time which could influence your results. Please explain your decision on this and potential bias introduced.

Reply: We agree with the concern. However, based on the data availability of USDA, only a small subset of sites recorded sowing and harvest dates. .

We have revised: "Unfortunately, only a small subset of sites recorded phenology data, and therefore trial-specific growing season lengths could not be used without omitting a large fraction of the data. The average plant and harvest dates in each site therefore were used based on Sacks et al. for both winter and spring wheat. A fixed time window was often set for some temperature accumulation indices in climate impact studies, rather than the growing season of each individual year. This is because the latter would result in endogeneity in an analysis of FDD, GDD and EDD on yields (e.g. warmer season may not have higher value of GDD as shorter growing season)." in lines 345-352.

Despite the endogeneity problem, we can still test whether this will influence our results. We randomly selected the planting and harvest dates in different years for a certain site based on the calendar range reported in Sacks et al. And we repeat the process for 1000 times, and did the regression analysis (Fig. R2). We found that:

(1) The yield impact by 1°C warming is slightly smaller than the results in Fig. 2. This reflects the endogeneity problem, and the degree-day cannot well reflect the actual temperature difference between years.

(2) The major conclusion in the main text still holds: the yield decline of MYG is smaller than CK for winter wheat, the results is contrary for spring wheat. Therefore, we believe crop calendar does not influence our major conclusion.

Fig. R2. Comparison of model estimates of yield response to 1°C warming using regression models based on the sites with randomly selected planting and harvest dates.

I am happy to review a revised version of the manuscript should it be required.

Good luck and kind regards,

Katharina Waha

Besides above major comments, we also revised based on your specific comments attached in PDF file. Please check our point-by-point reply as follows:

Line. 30: I suggest to give the relative yield decline here for elite varieties and check variety, instead of the relative difference to each other. The same for spring wheat. This way it is immediately clear what the expected changes is for each variety. But I leave that up to you, it is just an idea.

Reply: That is a great suggestion. Please check our revision: "Results shows that yields were declined by 3.6%/°C for elite winter wheat breeding lines, compared with -5.4%/°C when variety is held constant, reflecting a superior climate-resilience. However, advanced spring wheat breeding lines demonstrate 7.5% yield reduction per 1°C warming. That is more sensitive than the value of -7.1%/°C with constant variety planted, indicating an undermined climate-resilience for spring wheat." in lines of 32-37.

Line 41: remains→is

Reply: Revised as request. Please check line 45.

Line 42: I think you can simplify this and say 20% of protein and calories. Maybe also cite: B. Shiferaw, M. Smale, H.-J. Braun, E. Duveiller, M. Reynolds, G. Muricho, Crops that feed the world 10. Past successes and future challenges to the role played by wheat in global food security. Food Secur. 5, 291–317 (2013).

Reply: Revised as request and cite the new reference. Please check line 46 and line 400-402.

Line 72: our → this

Reply: Revised as request. Please check line 77.

Line 72: Can you please rewrite that sentence, it is not very clear. It is also a very long sentence, so perhaps split it into two. The information about the check varieties is not critical here so perhaps leave

that out. It should be explained in the figures and method section.

Reply: Revised as request. Please check: " The uniqueness of this dataset is that a constant check variety and an annual set of advanced breeding lines were planted together in the same trial each season. The variety Kharkof was used as the long-term winter wheat check, and the variety Marquis as the long-term spring wheat check. The two check varieties were planted in each site throughout the study period, which could be viewed as yield variability without variety replacement. On the other hand, the advanced breeding lines entered each year were different from the previous year, which reflects the ongoing nature of wheat breeding itself over the study period. Therefore, the difference in yield sensitivity (advanced breeding lines vs. check variety) can be defined as effectiveness of the current variety development under climate change, providing a footprint of the actual wheat breeding effort. " in lines of 77-87 and lines 324-325 in method section.

Fig. 3: I would show these results as boxplots for yield and temperature change over a period of 10 or 20 years, same as in Figure 4. I cannot see the need to show a time series plot here. But this is up to you so please disregard my comment if you do not agree.

Reply: Thanks for your suggestion, Fig. 3 was drawn as you suggested in the very first version. But, finally, we drew Fig. 3 like the current version because (1) the figure can tell us the key time point and temperature thresholds that MYG reduce to the historical CK in yield, we think this is important results to show to readers; (2) we want to make our figure format more diverse in the manuscript.

Fig. 4: I am not sure to understand the results shown in Figure 4. In line 127-128 you said that breeding does not provide an advantage in improving climate resilience because yield declines in eg. MYK are slightly stronger than for CK. Why is there a yield gain then in the results for future climate?

Reply: This is because the reference is different between Fig. 3 and Fig. 4. In Fig. 3, the yield changes is relative to the CK varieties at the baseline climate. This is to show the potential yield trend induced by climate compared to the historical yield level of CK .

However, the Fig. 4 is to show the yield gains of advanced breeding lines relative to the CK variety in the same future time periods. The Fig. 3h and Fig. 4b are actually consistent as can be confirmed by the smaller gap over time between MYG to CK in Fig. 3h.

We have revised the figure caption: " Figure 3. Model projection of wheat yield changes due to changes in climate of four categories of genotype under four SSPs, relative to the yield of CK at the baseline period." (lines 197-199). And " Figure 4. Percentage changes in yield gains of four categories of genotype and CK in same three future periods under four SSPs, projected by model. Boxplots show the ensemble range across six climate models (ACCESS-ESM1-5, BCC-CSM2-MR, CNRM-CM6-1, CNRM-ESM2-1, GFDL-ESM4, IPSL-CM6A-LR). Within a column of a SSP, LYG is yellow-bordered boxplot, MYG is blue-bordered boxplot, and HYG is red-bordered boxplot." (lines 216-221).

Line 206: The discussion section needs improvement. The text from L225 to L261 for example is about results for winter and spring wheat and partly repeating results. It is also too long. I suggest you avoid repeating results, shorten the discussion section, do not refer to earlier Figures and Tables. With almost 7 pages of discussion it is also worth using sub-headings or if not possible very clear introductory sentences for each paragraph.

Reply: We have moved some part of discussion to result part (lines 108-114; lines 223-241) and only focus on comparison with other's studies.

Line 216: add effect

Reply: We add "effect". Please check line 245.

Line 217-218: This should be moved to the results section.

Reply: Revised as request. Please check lines 108-114.

Line 225: Unclear to me. Do you mean: "Comparison between winter and spring wheat shows that..."

Reply: Sorry about the confusion. We have removed the sentence and now it is: " Warming poses a more harmful effect on yield of spring wheat breeding, resulting primarily from greater sensitivity to EDD of the advanced spring wheat breeding lines than CK. Spring wheat typically is at floral stage during a higher temperature cycle than winter wheat during its floral stage. " in lines 251-254.

Line 225: add "the"

Reply: We have removed the sentence and now it is: " Warming poses a more harmful effect on yield of spring wheat breeding, resulting primarily from greater sensitivity to EDD of the advanced spring wheat breeding lines than CK. Spring wheat typically is at floral stage during a higher temperature cycle than winter wheat during its floral stage. " in lines 251-254.

Line 225: add "variety"

Reply: We have removed the sentence and now it is: " Warming poses a more harmful effect on yield of spring wheat breeding, resulting primarily from greater sensitivity to EDD of the advanced spring wheat breeding lines than CK. Spring wheat typically is at floral stage during a higher temperature cycle than winter wheat during its floral stage. " in lines 251-254.

Line 334-335: I would like to know about management used. How much irrigation, fertilizer, pesticides were used? Did that change every year? Because it would be another reason for changes in yield over time.

Reply: The agronomic management information is not available to the published data of USDA. So, we add site-year terms in panel data model to remove the effects of non-climate factors (line 360). In our manuscript, we focus on yield changes due to climate, rather than non-climate factors.

Line 341- 343: Can you give examples of genotypes that typically yield lowest and highest for winter and for spring wheat? Or does that vary with time and location?

Reply: We can give examples of genotypes that yield lowest and highest in certain years but we cannot call this "typical" lowest or highest because the advanced breeding lines entered each year are different than the previous year in our breeding program. So this varies with time and location.

Line: 346-347: Do you have station data for each of the 92 sites or did you have to work with the data from the closest GHCN-D station? If the latter, please give the station names/station IDs used.

Reply: We add supplemental Table S5-S7, as request.

Line 358-359: Why did you use the Sacks et al. planting and harvest dates? I had a look at the 2021 report for spring wheat and the sowing and harvest dates for each location are given in the report. As

there will be deviations from the Sacks et al. data it would be much better to use the actual growing season data from the experiments. Also using Sacks et al dates would mean to keep the planting and harvest dates constant over time which might not have been the case in the experiments. If you must use the dates from elsewhere, can you please give an indication of the deviation from reported sowing and planting dates? And please discuss how that potential deviation influences the results.

Reply: For spring wheat, sowing and harvest dates were only documented since 1990s. In 1980s, some sites only report sowing and no harvest date or not reported for both dates. And in 1960s and 1970s, the sites that reported sowing and harvest dates are even less. For example, there is no data reported in 1973. Report just wrote that: " Planting and harvest dates were 1 to 2 weeks earlier than normal at most locations." Please check the link:

<https://wheat.pw.usda.gov/ggpages/gopher/Performance/hrswregional/Uniform%20Regional%20Hard%20Red%20Spring%20Wheat%20Nursery/1973/Statistics%20and%20Growing%20Conditions.txt.html>

For winter wheat, there is no sowing and harvest dates reported in 2020. Please check the link:

<https://www.ars.usda.gov/ARSUserFiles/30421000/HardWinterWheatRegionalNurseryProgram/2020%20SRPN%20021422%201429.xlsx>

We have revised: "Unfortunately, only a small subset of sites recorded phenology data, and therefore trial-specific growing season lengths could not be used without omitting a large fraction of the data. The average plant and harvest dates in each site therefore were used based on Sacks et al. for both winter and spring wheat. A fixed time window was often set for some temperature accumulation indices in climate impact studies, rather than the growing season of each individual year. This is because the latter would result in endogeneity in an analysis of FDD, GDD and EDD on yields (e.g. warmer season may not have higher value of GDD as shorter growing season)." in lines 345-352.

Despite the endogeneity problem, we can still test whether this will influence our results. We randomly selected the planting and harvest dates in different years for a certain site based on the calendar range reported in Sacks et al. And we repeat the process for 1000 times, and did the regression analysis (Fig. R2). We found that:

(1) The yield impact by 1°C warming is slightly smaller than the results in Fig. 2. This reflects the endogeneity problem, and the degree-day cannot well reflect the actual temperature difference between years.

(2) The major conclusion in the main text still holds: the yield decline of MYG is smaller than CK for winter wheat, the results is contrary for spring wheat. Therefore, we believe crop calendar does not influence our major conclusion.

Fig. R2. Comparison of model estimates of yield response to 1°C warming using regression models based on the sites with randomly selected planting and harvest dates.

Line 495: The link did not work when I tried it (April 2022). Please replace. Even better would be to have the data in a data repository with a permalink and DOI.

Reply: We have checked the Internet links of winter wheat nurseries. The link works from my place.

Please check it below:

<https://www.ars.usda.gov/plains-area/lincoln-ne/wheat-sorghum-and-forage-research/docs/hard-winter-wheat-regional-nursery-program/research/>

The data belongs to USDA. And is already publically available online. See the links to the data that are provided.

Reviewers' Comments:

Reviewer #1:

Remarks to the Author:

Reviewer Blind Comments to Author

The authors have adequately addressed my comments and those from the other two reviewers. A main weakness of the article is the exclusion of CO₂ fertilization in regression models of historic yields and subsequent yield projection into year 2100 under future climate.

I have a few minor comments related to the revision of the manuscripts listed below.

Specific comments

L32: Results shows  Results show

L33: 3.6%/oC  3.6% per 1 °C warming

L33-34: compared with -5.4%/°C when variety is held constant, reflecting a superior climate-resilience  compared with -5.4% for the check variety, indicating a superior climate-resilience

L35: demonstrate 7.5% yield reduction per 1 °C warming. That is more sensitive than the value of -7.1%/oC with constant variety planted, indicating an undermined climate-resilience for spring wheat.  showed a 7.5% yield reduction per 1 °C warming, which is more sensitive than a 7.1% reduction for the check variety, indicating less climate-resilience for spring wheat.

L40-41: Our study highlights that following the current wheat breeding adaptation progress is challenging to abate climate warming  How about this: Our study highlights that the adaptation progress following the current wheat breeding strategies is challenging to abating climate warming.

L59: this mechanism  these negative effects

L69-70: field observations enabling comparison between yield sensitivities  field observations, hindering comparison in yield sensitivities

L77 : a constant check  a common check

L111: Increase in FDD  You mean decrease in FDD?

L143: by 7.5% yield decline  with 7.5% yield decline

L148-166: -> These might fit better to the Discussion section.

L151: was repeated but used the  was repeated using the

L165-166: is not relevant to  How about: is insensitive to

L176: is mixed in direction of changes  is mixed in the direction of changes

L213: projected to be reduce to 43.8%  projected to be reduced to 43.8%

L215: Figure 4 legend: missing HYG legend

L215: Figure 4 legend: Incorrect name for different SPPs

L216: Figure 4 caption: there is no need to list the six climate models

L232: heading phases have shifted to an earlier date over time  Note: This could be due to shifting of planting to earlier dates.

L223-241:  These might fit better to the Discussion section.

L253: Spring wheat typically is at floral stage during a higher temperature cycle than winter wheat during its floral stage  The floral stage of spring wheat typically coincides with a higher temperature cycle than winter wheat

L256: indicating winter wheat is adapting to climate warming in a positive manner  indicating greater climate adaptability.

L257-258: Such contrasting responses result in yield benefits due to variety breeding increases in winter wheat and decreases in spring wheat  Such contrasting responses result in greater yield benefits for winter wheat than for spring wheat from breeding progress.

L261: warming is more different and challenging  warming is more challenging

L288: Northward shifting area of winter wheat  Northward shifting of winter wheat production

L292: are required to be implement now  are required to be implement now

L300-303: our study presents for the first time within a 59-year-long and multi-site dataset from breeding nurseries in which yield responses of new wheat varieties to climate were compared with long-term constant CK varieties in each season under rainfed conditions in North America  our study presented a first comprehensive analysis, using a multi-year and multi-site dataset from breeding nurseries, on yield responses of new wheat varieties to climate under rainfed conditions in North America

L305-306: offset yield benefits by switching to new varieties in the assumption that the current breeding progress would continue  offset yield gains with the continuation of current breeding progress

L308-311: Innovation on breeding technologies accompanied with alternation in traditional breeding processes preparing for more extreme climate can help to ensure the future productivity and climate resilience of wheat in a changing climate  Integration of innovative technologies with traditional approaches in breeding for future climate can help to ensure the future productivity and climate resilience of wheat in a changing climate.

Reviewer #4:

Remarks to the Author:

The response to the review comments is thorough and very much satisfactory. The only additional minor quibble that I have is over the assertion in the abstract that the slightly higher estimated yield reduction from the advanced breed spring wheat implies that climate resilience is "undermined" by the breeding. I'm not convinced from the analysis that this represents a statistically significant difference so I suspect it would be more correct to say that "climate resilience is not improved and may even decline" or similar language.

Responses to reviewers' comments

We highly appreciate the insightful comments that improve the quality of the manuscript. We have revised the language mistakes following these suggestions and we also ask a native English editor polish the language. Please check our point-by-point responses. Our reply has been highlighted by blue.

Reviewer #1 (Remarks to the Author):

The authors have adequately addressed my comments and those from the other two reviewers. A main weakness of the article is the exclusion of CO₂ fertilization in regression models of historic yields and subsequent yield projection into year 2100 under future climate.

Reply: We greatly appreciate your insightful suggestions, and agree with the weakness you said. We address the weakness in Introduction (line 95-97) and Discussion section (line 271-280).

I have a few minor comments related to the revision of the manuscripts listed below.

Specific comments

L32: Results shows  Results show

Reply: We revised. Please check line 38.

L33: 3.6%/oC  3.6% per 1°C warming

Reply: We revised. Please check line 38.

L33-34: compared with -5.5%/°C when variety is held constant, reflecting a superior climate-resilience  compared with -5.5% for the check variety, indicating a superior climate-resilience

Reply: We revised. Please check line 39-40.

L35: demonstrate 7.5% yield reduction per 1°C warming. That is more sensitive than the value of -7.1%/oC with constant variety planted, indicating an undermined climate-resilience for spring wheat.  showed a 7.5% yield reduction per 1°C warming, which is more sensitive than a 7.1% reduction for the check variety, indicating less climate-resilience for spring wheat.

Reply: We revised. Please check line 41-43. We slightly changed your revision to follow the reviewer #4.

L40-41: Our study highlights that following the current wheat breeding adaptation progress is challenging to abate climate warming  How about this: Our study highlights that the adaptation progress following the current wheat breeding strategies is challenging to abating climate warming.

Reply: We revised. Please check L. 46-47.

L59: this mechanism  these negative effects

Reply: We revised. Please check line 62.

L69-70: field observations enabling comparison between yield sensitivities  field observations, hindering comparison in yield sensitivities

Reply: We revised, and English editor suggests a clear way for the sentence. Please check line 72-73.

L77 : a constant check  a common check

Reply: We revised. Please check line 79.

L111: Increase in FDD  You mean decrease in FDD?

Reply: The regression coefficients is negative, indicating less yield with higher FDD. This is because higher FDD indicates more serious freezing temperature.

L143: by 7.5% yield decline  with 7.5% yield decline

Reply: We revised. Please check line 134.

L148-166: -> These might fit better to the Discussion section.

Reply: We moved it to Discussion section. Please check line 181-202.

L151: was repeated but used the  was repeated using the

Reply: We revised. Please check line 185.

L165-166: is not relevant to  How about: is insensitive to

Reply: We revised. Please check line 201.

L176: is mixed in direction of changes  is mixed in the direction of changes

Reply: We revised. Please check line 149.

L213: projected to be reduce to 43.8%  projected to be reduced to 43.8%

Reply: We revised. Please check line 173.

L215: Figure 4 legend: missing HYG legend

Reply: We have revised the legend of HYG in Figure 4. Please check the figure.

L215: Figure 4 legend: Incorrect name for different SPPs

Reply: We have revised the legend of SSPs in Figure 4. Please check the figure.

L216: Figure 4 caption: there is no need to list the six climate models

Reply: We have removed the names. Please check the line 541-548.

L232: heading phases have shifted to an earlier date over time  Note: This could be

due to shifting of planting to earlier dates.

Reply: We agree with that. We rechecked that reference but they do not show us the time trends of day after sowing. Therefore, we removed this sentence.

L223-241:  These might fit better to the Discussion section.

Reply: We moved it to Discussion section. Please check line 222-238.

L253: Spring wheat typically is at floral stage during a higher temperature cycle than winter wheat during its floral stage  The floral stage of spring wheat typically coincides with a higher temperature cycle than winter wheat

Reply: We revised. Please check line 209-210.

L256: indicating winter wheat is adapting to climate warming in a positive manner  indicating greater climate adaptability.

Reply: We revised. Please check line 212.

L257-258: Such contrasting responses result in yield benefits due to variety breeding increases in winter wheat and decreases in spring wheat  Such contrasting responses result in greater yield benefits for winter wheat than for spring wheat from breeding progress.

Reply: We revised. Please check line 212-213.

L261: warming is more different and challenging  warming is more challenging

Reply: We revised. Please check line 216.

L288: Northward shifting area of winter wheat  Northward shifting of winter wheat production

Reply: We revised. Please check line 259.

L292: are required to be implement now  are required to be implement now

Reply: English editor changes the whole structure of the sentence. Please check line 263-265.

L300-303: our study presents for the first time within a 59-year-long and multi-site dataset from breeding nurseries in which yield responses of new wheat varieties to climate were compared with long-term constant CK varieties in each season under rainfed conditions in North America  our study presented a first comprehensive analysis, using a multi-year and multi-site dataset from breeding nurseries, on yield responses of new wheat varieties to climate under rainfed conditions in North America

Reply: We revised. We also remove "first" as request by editor. Please check line 281-283.

L305-306: offset yield benefits by switching to new varieties in the assumption that

the current breeding progress would continue  offset yield gains with the continuation of current breeding progress

Reply: We revised. Please check line 285-286.

L308-311: Innovation on breeding technologies accompanied with alternation in traditional breeding processes preparing for more extreme climate can help to ensure the future productivity and climate resilience of wheat in a changing climate  Integration of innovative technologies with traditional approaches in breeding for future climate can help to ensure the future productivity and climate resilience of wheat in a changing climate.

Reply: We revised. Please check line 288-290.

Reviewer #4 (Remarks to the Author):

The response to the review comments is thorough and very much satisfactory. The only additional minor quibble that I have is over the assertion in the abstract that the slightly higher estimated yield reduction from the advanced breed spring wheat implies that climate resilience is "undermined" by the breeding. I'm not convinced from the analysis that this represents a statistically significant difference so I suspect it would be more correct to say that "climate resilience is not improved and may even decline" or similar language.

Reply: We greatly appreciate your suggestion and we revised the sentence. Please check line 42-43.